# EFFICIENT DISCOVERY OF DYNAMICAL LAWS IN SYMBOLIC FORM

## ABSTRACT

We propose a transformer-based sequence-to-sequence model that recovers scalar ordinary differential equations (ODEs) in symbolic form from time-series data of a single observed solution trajectory of the ODE. Our method is efficiently scalable: after one-time pretraining on a large set of ODEs, we can infer the governing laws of a new observed solution in a few forward passes of the model. First, we generate and make available a large dataset of more than 3M ODEs together with more than 63M numerical solutions for different initial conditions that may serve as a useful benchmark for future work on machine learning for dynamical systems. Then we show that our model performs better or on par with existing methods in various test cases in terms of accurate symbolic recovery of the ODE, especially for more complex expressions. Reliably recovering the symbolic form of dynamical laws is important as it allows for further dissemination of the inferred dynamics as well as meaningful modifications for predictions under interventions.

## 1 INTRODUCTION

*Science* is commonly described as the "discovery of natural laws through experimentation and observation". Researchers in the natural sciences increasingly turn to machine learning (ML) to aid the discovery of natural laws from observational data alone, which is often abundantly available, hoping to bypass expensive and cumbersome targeted experimentation. While there may be fundamental limitations to what can be extracted from observations alone, recent successes of ML in the entire range of natural sciences provide ample reason for excitement. In this work, we focus on ordinary differential equations, a ubiquitous description of dynamical natural laws in physics, chemistry, and systems biology. For a first order ODE $\dot{y} := \partial y/\partial t = f(y, t)$, we call $f$ (which uniquely defines the ODE) the underlying dynamical law. Informally, our goal is then to infer $f$ in symbolic form given discrete time-series observations of a single solution $\{y_i := y(t_i)\}_{i=1}^n$ of the underlying ODE.

Contrary to "black-box-techniques" such as Neural Ordinary Differential Equations (NODE) (Chen et al., 2018) that aim at inferring a possible $f$ as an arguably opaque neural network, we focus specifically on symbolic regression. From the perspective of the sciences, a law of nature is useful insofar as it is more broadly applicable than to merely describe a single observation. In particular, the reason to learn a dynamical law in the first place is to dissect and understand it as well as to make predictions about situations that differ from the observed one. From this perspective, a symbolic representation of the law (in our case the function $f$) has several advantages over block-box representations: they are compact and directly interpretable, they are amenable to analytic analysis, they allow for meaningful changes and thus enable assessment of interventions and counterfactuals.

In this work, we develop **N**eural **S**ymbolic **O**rdinary **D**ifferential **E**quation (**NSODE**), a sequence-to-sequence transformer to efficiently infer governing ODEs in symbolic form from a single observed solution trajectory that makes use of massive pretraining. We first (randomly) generate a total of >3M scalar, autonomous, non-linear, first-order ODEs, together with a total of >63M numerical solutions from various (random) initial conditions. All solutions are carefully checked for convergence of the numerical integration. This dataset is unprecedented in both its scale and diversity and will be made publicly available alongside the code that was used to generate it.

We then devise NSODE, a sequence-to-sequence transformer that maps observed trajectories, i.e., numeric sequences of the form $\{(t_i, y_i)\}_{i=1}^n$, directly to symbolic equations as strings, e.g., `"y**2+1.64*cos(y)"`, which is the prediction for $f$. This example directly highlights the benefit

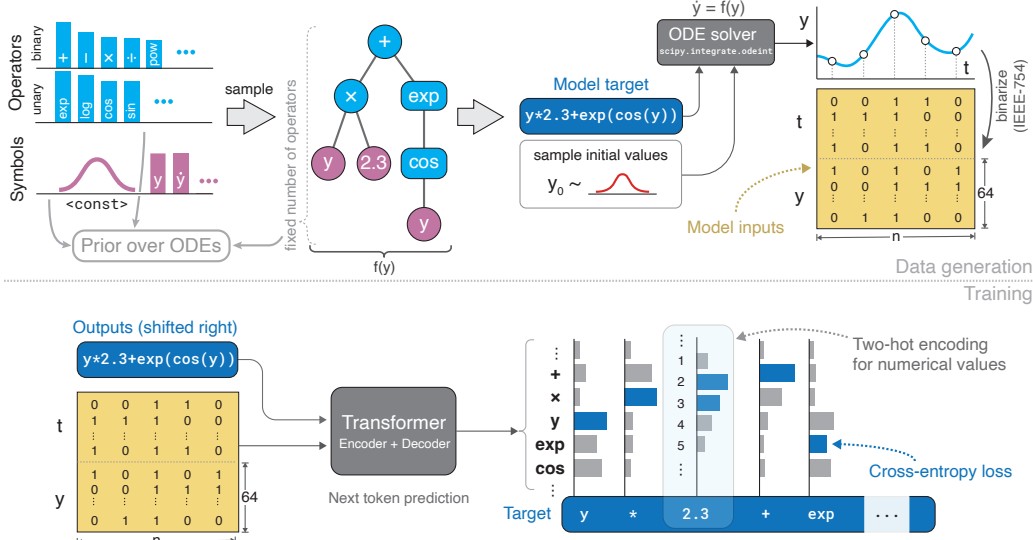

Figure 1: An overview illustration of the data generation (top) and training pipeline (bottom). Our dataset stores solutions in numerical (non-binarized) form on the entire regular solution time grid.

of symbolic representations in that the $y^2$ and $\cos(y)$ terms tell us something about the fundamental dynamics of the observed system; the constant `1.64` will have semantic meaning in a given context and we can, for example, make predictions about settings in which this constant takes a different value. NSODE combines and innovates on technical advances regarding input representations and an efficiently optimizable loss formulation. Our model outperforms scalable existing methods in terms of skeleton recovery, especially on more complex expressions. While other methods still perform better on simple expressions found in (adjusted) existing benchmark datasets (which these methods have been tuned to) as well as a novel set of simple ODEs that we manually collected from different domains, they are typically orders of magnitude slower than NSODE.

## 2 BACKGROUND AND RELATED WORK

Modeling dynamics and forecasting their behavior has a long history in machine learning. While NODEs (Chen et al., 2018) (with a large body of follow up work) are perhaps the most prominent approach, their inherent black-box character complicates scientific understanding of the observed phenomena. Recent alternatives such as tractable dendritic RNNs (Brenner et al., 2022) and Neural Operators (Kovachki et al., 2021; Li et al., 2021) set out to facilitate scientific discovery by combining accurate predictions with improved interpretability. A considerable advantage of these and similar approaches is their scalability to high dimensional systems as well as their relative robustness to noise, missing or irregularly sampled data (Iakovlev et al., 2021) and challenging properties such as multiscale dynamics (Vlachas et al., 2022) or chaos (Park et al., 2022; Patel & Ott, 2022).

Despite these advantages, we turn the focus to a different class of models in this paper and look at approaches that explicitly predict a mathematical expression in symbolic form that describes the observed dynamics. Such models are in a sense situated on the other side of the spectrum: while less scalable, the predicted symbolic expression is compact and readily interpretable so that dynamical properties can be analytically deduced and investigated. A recent benchmark study and great overview including deep learning-based as well as symbolic models can be found in Gilpin (2021).

**Evolutionary algorithms.** Classically, symbolic regression is approached through evolutionary algorithms such as genetic programming (Koza, 1993). Genetic programming randomly evolves a population of prospective mathematical expressions over many iterations and mimics natural selection by keeping only the best contenders across iterations, where superiority is measured by user-defined fitness functions (Schmidt & Lipson, 2009). Process-based modeling follows a similar approach but includes domain knowledge-informed constraints on particular components of the system in order to reduce the search space to reasonable candidates (Todorovski & Dzeroski, 1997; Bridewell et al., 2008; Simidjievski et al., 2020).

**Gradient-based search.** More recently, symbolic regression has been approached with machine learning methods which exploit gradient information to optimize within the space of (finite) compositions of pre-defined basis functions. Brunton et al. (2016) builds on sparse linear regression to identify a linear combination of basis functions that yields the best fit for the observed data. This approach has inspired a large body of follow-up work that generalize the idea to partial observations (Bakarji et al., 2022), parameterized functions (Lejarza & Baldea, 2022), simultaneous discovery of coordinates (Champion et al., 2019), coordinate transformations that linearize the dynamics (Lusch et al., 2018) and partial differential equations (Rudy et al., 2017). These techniques often deploy sparsity-promoting regularizers and train one model for each set of observations. Once trained, the model itself represents the predicted symbolic expression, which can be read off the non-zero coefficients. This modeling principle is also employed by many other approaches that replace linear regression by neural networks with diverse sets of activation functions, both for differential equations (Long et al., 2019; Liu et al., 2020) and non-differential algebraic equations (Sahoo et al., 2018).

**Hybrid models.** Alternatively, one can train a model to directly output the symbolic expressions (for example as a string). Supervised learning with gradient-based optimization for this approach is challenged by the formulation of a differentiable loss that measures the fit between the predicted symbolic expression and the observed data. Thus, prior work, mostly in the context of non-differential equation prediction, resorted to reinforcement learning (Petersen et al., 2021) or combinations of neural networks and evolutionary algorithms (Atkinson et al., 2019; Costa et al., 2021). A hybrid approach combining gradient-free, human intuition-guided heuristic search via genetic programming with neural network-based optimization has been presented for non-differential equations by Udrescu et al. (2020) and extended to dynamical systems by Weilbach et al. (2021).

**Sequence-to-sequence models.** The closest works to ours use pre-trained, attention-based sequence-to-sequence models for symbolic regression *of functional relationships* (Biggio et al., 2021; Valipour et al., 2021; Kamienny et al., 2022; Vastl et al., 2022) or (discrete) recurrence relations (D'Ascoli et al., 2022). They exploit the fact that symbolic expressions for (multi-variate) scalar functions can be both generated and evaluated on random inputs cheaply, resulting in essentially unlimited training data. Large data including ground-truth expressions in symbolic form allow for a differentiable cross-entropy loss based directly on the symbols of the expression, instead of the numerical proximity between evaluations of the predicted and true expression. While the cross-entropy loss works well for operators and symbols (e.g. `+`, `exp`, `sin`, `x`, `y`), a naive implementation is inefficient for numerical constants, e.g. `1.452`. As a workaround previous works resort to one of two strategies: 1) represent all constants with a special `<const>` token when training the sequence-to-sequence model and predict only the presence of a constant. Actual values are then inferred in a second, subsequent optimization step that minimizes $\sum_{i=1}^{n} \|y_i - \hat{y}_{c_1,c_2,...}(x_i)\|^2$. This second optimization procedure comes with substantial computational cost as constants have to be fit per inferred expression. In particular, we highlight that it does not transfer to inferring ODEs as it would require to first solve the predicted ODE $\dot{y} = \hat{f}(y)$ to obtain predicted $\{\hat{y}_i\}_{i=1}^n$ values that can be compared to the set of observations $\{y_i\}_{i=1}^n$. While differentiable ODE solvers exist, optimizing constants this way is prohibitively expensive and typically highly unstable. 2) A second popular strategy consists in rounding constants within the range of interest so that they can be represented with a finite number of tokens. This second strategy avoids a subsequent optimization step and enjoys clever encoding schemes with improved token efficiency yet represents values with an inherent loss in precision. As an alternative, we develop a representation based on a "two-hot" encoding which avoids subsequent optimization steps as well as rounding.

We note that many previous symbolic regression methods have been described as "discovering natural laws". However, most of them learn *fixed functional relationships* from input-output pairs, whereas we seek to actually infer *the underlying dynamic law* that governed the behavior of the observed solution trajectory. A common obstacle to inferring dynamical laws is the lack of temporal derivatives in empirical measurements, which can hence not serve as regression targets. While derivatives can be estimated numerically, for example via finite differences, this requires dense, regularly sampled observations with a high signal-to-noise ratio to prevent fatal propagation of approximation errors. Alternative loss formulations that bypass unobserved time derivatives have been identified as an open challenge in symbolic regression for dynamical systems recently (Qian et al., 2022). With NSODE, we propose a solution that does not rely on temporal derivatives and thus avoids these complications.

## 3   METHOD

**Problem setting.**  Given observations $\{(t_i, y_i)\}_{i=1}^n$ of a trajectory $y : [t_1, t_n] \to \mathbb{R}$ that is a solution of the ODE $\dot{y} = f(y)$, we aim to recover the function $f$ in symbolic form—in our case as a string. In this work, we focus on time-invariant (or autonomous) ODEs (i.e., $f(y, t) = f(y)$). Such settings are a good starting point for investigation as they are commonly studied and can be thought of as "evolving on their own" without external driving forces or controls, i.e., once an initial condition is fixed the absolute time does not directly influence the evolution. We explicitly assume that the observed system actually evolves according to an ODE in canonical form $\dot{y} = f(y)$ such that $f$ can be expressed in closed form using the mathematical operators seen during training (see Section 3.1). To sum up, in this work we study the rich class of non-linear, scalar, first-order, autonomous ODEs, even though our approach is not fundamentally limited to this setting. In Appendix A, we discuss extensions of NSODE to higher-order systems of coupled non-autonomous ODEs.

### 3.1   DATA GENERATION

**Sampling symbolic expressions.**     To exploit large-scale supervised pretraining we randomly generate a dataset of $\sim$63M ODEs in symbolic form along with their numerical solutions for randomly sampled initial values. Since we assume ODEs to be in canonical form $\dot{y} = f(y)$, generating an ODE is equivalent to generating a symbolic expression $f(y)$. We follow Lample & Charton (2019), who sample such an expression $f(y)$ as a unary-binary tree, where each internal node corresponds to an operator and each leaf node corresponds to a constant or variable. The algorithm consists of two phases: (1) A unary-binary tree is sampled uniformly from the distribution of unary-binary trees with up to $k \in \mathbb{N}$ internal nodes, which crucially does not overrepresent small trees corresponding to short expressions. Here the maximum number of internal nodes $K$ is a hyperparameter of the algorithm. (2) The sampled tree is "decorated", that is, each binary internal node is assigned a binary operator, each unary internal node is assigned a unary operator, and each leaf is assigned a variable or constant. Hence, we have to specify a distribution over the $N_{\mathrm{bin}}$ binary operators, one over the $N_{\mathrm{una}}$ unary operators, a probability $p_{\mathrm{sym}} \in (0, 1)$ to decide between symbols and constants, as well as a distribution $p_{\mathrm{c}}$ over constants. For constants we distinguish explicitly between sampling an integer or a real value. Together with $K$, these choices uniquely determine a distribution over equations $f$ and are described in detail in Appendix B. Figure 1 depicts an overview of the data generation procedure.

The pre-order traversal of a sampled tree results in the symbolic expression for $f$ in prefix notation. After conversion to the more common mathematical infix notation, we simplify each expression using the computer algebra system SymPy (Meurer et al., 2017), and filter out constant equations $f(y) = c$ as well as expressions that contain operators or symbols that were not part of the original distribution[1]. We call the structure modulo the value of the constants of such an expression a **skeleton**. Any skeleton containing at least one binary operator or constant can be represented by different unary-binary trees. Vice versa many of the generated trees will be simplified to the same skeleton. To ensure diversity and to mitigate potential dataset bias towards particular expressions, we discard duplicates on the skeleton level. To further cheaply increase the variability of ODEs we sample $N_{\mathrm{const}}$ unique sets of constants per skeleton. When sampling constants we take care not to modify the canonical expression by adhering to the rules listed in Appendix B.1. Our final dataset contains linear and non-linear as well as homogeneous and inhomogeneous ODEs and we provide summary statistics about the distribution over equations in Appendix D. Besides the number of internal nodes, we also report a simple yet common measure of **complexity** for each symbolic equation, which is the overall count of symbols (e.g., $y$, or constants) as well as operators in an expression.

**Computing numerical solutions.**  We obtain numerical solutions for all ODEs via SciPy's interface (Virtanen et al., 2020) to the LSODA software package (Hindmarsh & Laboratory, 1982) with both relative and absolute tolerances set to $10^{-9}$. LSODA consists of a collection of ODE solvers and implements a strategy to automatically choose an appropriate solver for the problem at hand (e.g., recognizing stiff problems). We solve each equation on a fixed time interval $t \in [0, T]$ and store solutions on a regular grid of $N_{\mathrm{grid}}$ points. For each ODE, we sample up to $N_{\mathrm{iv}}$ initial values $y(0) = y_0$ uniformly from $(y_0^{\min}, y_0^{\max})$.[2] While LSODA attempts to select an appropriate solver, numerical solutions still cannot be trusted in all cases. Therefore, we check the validity of solutions

---

[1]With the exception of a unary $-$, which we do not discard.

[2]Due to a timeout per ODE, fewer solutions may remain if the solver fails for repeated initial value samples.

Table 1: Overview of our model architecture.

| | Encoder | Decoder |
|---|---|---|
| architecture | BigBird[†] | BigBird |
| layers | 6 | 6 |
| heads | 16 | 16 |
| embed. dim. | 512 | 512 |
| forward dim. | 2048 | 2048 |
| activation | gelu | gelu |
| vocab. size | - | 43 |
| position enc. | learned | learned |
| parameters | 23.3M | 23.3M |

[†]We use full attention and chose BigBird (Zaheer et al., 2020) for its fast Huggingface implementation.

via the following quality control check: we use 9th order central finite differences to approximate the temporal derivative of the solution trajectory (on the same temporal grid as the proposed solution), denoted by $\dot{y}_{\text{fd}}$, and filter out any solution for which $\|\dot{y}_{\text{fd}} - \dot{y}\|_\infty > \epsilon$, where we use $\epsilon = 1$.

## 3.2 MODEL DESIGN CHOICES

NSODE consists of an encoder-decoder transformer with architecture choices listed in Table 1. We provide a visual overview in Figure 1.

**Representing input trajectories.** A key difficulty in feeding numerical solutions $\{y_i\}_{i=1}^n$ as input is that their range may differ greatly both within a single solution as well as across ODEs. For example, the linear ODE $\dot{y} = c \cdot y$ for a constant $c$ is solved by an exponential $y(t) = y_0 \exp(ct)$ for initial value $y(0) = y_0$, which may span many orders of magnitude on a fixed time interval. To prevent numerical errors and vanishing or exploding gradients caused by the large range of values, we assume each representable 64-bit float value is a token and use its IEEE-754 encoding as the token representation (Biggio et al., 2021). We thus convert all pairs $(t_i, y_i)$ to their IEEE-754 64 bit representations, channel them through a linear embedding layer, and then feed them to the encoder.

**Representing symbolic expressions.** The target sequence (i.e., the string for the symbolic expression of $f$) is tokenized on the symbol-level. We distinguish two cases: (1) *Operators and variables:* for each operator and variable we include a unique token in the vocabulary. These tokens are one-hot encoded and passed through a learnable embedding layer before their embedded representations are fed to the decoder.(2) *Numerical constants:* constants may come from both discrete (integers) as well as continuous distributions, as for example in `y**2+1.64*cos(y)`. Hence, it is unfeasible to include individual tokens "for each constant". Naively tokenizing on the digit level, i.e., representing real values literally as the sequence of characters (e.g., `"1.64"`), not only significantly expands the length of target sequences and thus the computational cost, but also requires a variable number of prediction steps for every single constant.

Instead, we take inspiration from Schrittwieser et al. (2020) and encode constants in a *two-hot* fashion. We fix a finite homogeneous grid on the real numbers $x_1 < x_2 < \ldots < x_m$ for some $m \in \mathbb{N}$, which we add as tokens to the vocabulary. The grid range $(x_1, x_m)$ and number of grid points $m$ can be tuned for performance. Our choices are described in Appendix B.3. For any constant $c$ in the target sequence we then find $i \in \{1, \ldots, m-1\}$ such that $x_i \leq c < x_{i+1}$ and encode $c$ as a distribution supported on $x_i, x_{i+1}$ with weights $\alpha, \beta$ such that $\alpha x_i + \beta x_{i+1} = c$. That is, the target in the cross-entropy loss for a constant token is not a strict one-hot encoding, but a distribution supported on two (neighboring) vocabulary tokens resulting in a lossless encoding of continuous values in $[x_1, x_m]$. While this two-hot representation can be used directly in the cross-entropy loss function and thus greatly facilitates training, it can not be passed directly through an embedding layer. For a generic constant in the target sequence represented as $\alpha x_i + \beta x_{i+1}$, we thus instead provide the linear combination of the two embeddings $\alpha \, \texttt{embed(x\_i)} + \beta \, \texttt{embed(x\_{i+1})}$ as decoder input.

**Decoding constants.** When decoding a predicted sequence, we check at each prediction step whether the $\arg\max$ of the logits corresponds to one of the $m$ constant tokens $\{x_1, \ldots, x_m\}$. If not, we proceed by conventional 1-hot decoding to obtain predicted operators and variables. If instead the argmax corresponds to, for example, $x_i$, we also pick its largest-logit neighbor ($x_{i-1}$ or $x_{i+1}$; suppose $x_{i+1}$), renormalize their probabilities by applying a softmax to all logits and use the resulting two probability estimates as weights $\alpha, \beta$. Constants are then ultimately decoded as $\alpha x_i + \beta x_{i+1}$.

### 3.3 EVALUATION AND METRICS

**Sampling solutions.** To infer a symbolic expression for the governing ODE of a new observed solution trajectory $\{(t_i, y_i)\}_{i=1}^n$, all the typical policies such as greedy, sampling, or beam search are available. In our evaluation, we leverage computationally cheap forward passes to perform beam search with 1536 beams and report top-$k$ results with $k$ ranging from 1 to 1536. To pick from the candidate expressions provided by the beam search, one could for example only provide a fraction of the trajectory as input to the model (say the first half) and then pick the candidate ODE whose numerical solution best predicts the trajectory on the remaining observation.

**Metrics.** We evaluate model performance both numerically and symbolically. For numerical evaluation we follow Biggio et al. (2020): suppose the ground truth ODE is given by $\dot{y} = f(y)$ with (numerical) solution $y(t)$ and the predicted ODE is given by $\hat{\dot{y}} = \hat{f}(y)$. To compute numerical accuracy we first evaluate $f$ and $\hat{f}$ on $N_{\text{eval}}$ points in the interval $[\min(y(t)), \max(y(t))]$ (i.e., the interval traced out by the observed solution), which yields function evaluations $\texttt{gt} = \{\dot{y}_i\}_{i=1}^{N_{\text{eval}}}$ and $\texttt{pred} = \{\hat{\dot{y}}_i\}_{i=1}^{N_{\text{eval}}}$. We then assess whether $\texttt{numpy.allclose}$[3] returns $\texttt{True}$ as well as whether the coefficient of determination $\mathrm{R}^2 \geq 0.999$.[4] Numerical evaluations capture how closely the predicted function approximates the ground truth function within the interval $[\min(y(t)), \max(y(t))]$.

However, a key motivation for symbolic regression is to uncover a *symbolic* mathematical expression that governs the observations. Perhaps surprisingly, previous works on symbolic regression have paid little attention to directly testing for symbolic equivalence, which need not be implied by numerical fit. Testing for symbolic equivalence between ground truth expression $f(y)$ and a predicted expression $\hat{f}(y)$ is unsuitable in the presence of real-valued constants as even minor deviations between true and predicted constants render the equivalence false. Instead, we regard the predicted expression $\hat{f}(y)$ to be symbolically correct if $f(y)$ and $\hat{f}(y)$ can be made equivalent by modifying only the values of constants in the predicted expression $\hat{f}(y)$. This is implemented using SymPy's $\texttt{match}$ function. In order not to alter the structure of the predicted expression, we constrain modifications of constants such that all constants remain non-zero and retain their original sign. This definition is thus primarily concerned with the structure of an expression, rather than precise numerical agreement. Once the structure is known, the inference problem becomes conventional parameter estimation. We report percentages of samples in a given test set that satisfies any individual metric (numerical and symbolic), as well as percentages satisfying symbolic and numerical metrics simultaneously.

## 4 EXPERIMENTS

### 4.1 BENCHMARK DATASETS

We construct several test sets to evaluate model performance and generalization in different settings.

- **testset-iv**: Our first test set assesses generalization within initial values not seen during training. It consists of 5793 ODEs picked uniformly at random from our generated dataset but re-sampled initial values. We also employ the following constraints via rejection sampling: (a) All skeletons in testset-iv are unique. (b) As the number of unique skeletons increases with the number of operators, we allow at most 2000 examples per number of operators (with substantially fewer unique skeletons existing for few operators).
- **testset-constants**: Our second test set assesses generalization within unseen initial values and constants. It consists of 2720 ODEs picked uniformly at random from our dataset (ensuring unique skeletons and at most 1000 examples per number of operators as above), but re-sampled intial values and constants.
- **testset-skeletons**: In principle, we can train NSODE on all possible expressions (using only the specified operators and number ranges) up to a specified number of operators. However, even with the millions examples in our dataset, we have by far not exhausted the huge space of possible

---

[3] $\texttt{numpy.allclose}$ returns True if $\texttt{abs(a - b) <= (atol + rtol * abs(b))}$ holds element-wise for elements $a$ and $b$ from the two input arrays. We use $\texttt{atol=1e-10}$ and $\texttt{rtol=0.05}$; $a$ corresponds to predictions, $b$ corresponds to ground truth.

[4] For observations $y_i$ and predictions $\hat{y}_i$ we have $\mathrm{R}^2 = 1 - (\sum_i (y_i - \hat{y}_i)^2)/(\sum_i (y_i - \overline{y})^2)$.

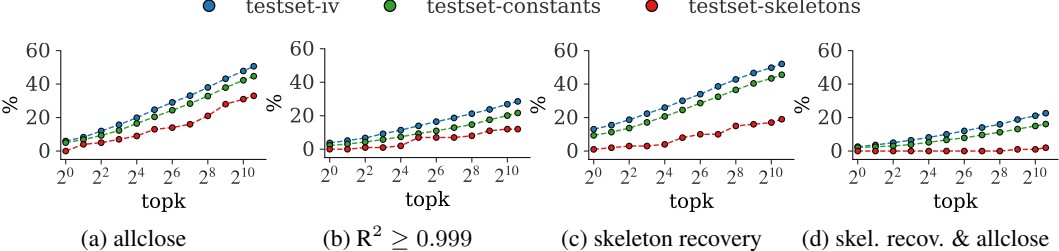

Figure 2: We evaluate numerical (allclose, $R^2$) and symbolic (skeleton recovery) metrics as well as combined skeleton recovery & allclose on testset-iv, testset-constants, and testset-skeletons as $k$ increases .

skeletons (especially for larger numbers of operators). Hence, our third test set contains 100 novel random ODEs with skeletons that were never seen during training.

- **testset-iv-163**: This is a subset of testset-iv motivated by the fact that most symbolic regression models we want to compare to require a separate optimization for every individual example, which was computationally infeasible for our testset-iv. For a fair comparison, we therefore subsampled up to 10 ODEs per complexity uniformly at random, yielding 163 examples.

- **testset-textbook**: To assess how NSODE performs on "real problems", we manually curated 12 scalar, non-linear ODEs from Wikipedia pages, physics textbooks, and lecture note from university courses on ODEs. These equations are listed in Table 7 in Appendix C. We note that they are all extremely simple compared to the expressions in our generated dataset in that they are ultimately mostly low order polynomials, some of which with one fractional exponent.

- **testset-classic**: To validate our approach on existing datasets we turn to benchmarks in the classic symbolic regression literature (inferring just the functional relationship between input-ouput pairs) and simply interpret functions as ODEs. In particular we include all scalar function listed in the overview in McDermott et al. (2012) which includes equations from many different benchmarks Keijzer (2003); Koza (1993; 1994); Uy et al. (2011); Vladislavleva et al. (2008). For example, we interpret the function $f(y) = y^3 + y^2 + y$ from Uy et al. (2011) as an autonomous ODE $\dot{y}(t) = f(y(t)) = y(t)^3 + y(t)^2 + y$, which we solve numerically for a randomly sampled initial value as described before.

## 4.2 BASELINES

We compare our method to recent popular baselines from the literature (see Section 2). We briefly describe them including some limitations here and defer all details to Appendix E. All baselines work by explicitly fitting a separate regression function per individual ODE to map the observed samples of the solution trajectory to their temporal derivatives $y(t) \mapsto \hat{y}(t)$, using the coefficient of determination $R^2$ as optimization objective. Since derivatives $\hat{y}(t)$ are typically not observed, we approximate them via finite differences using the implementation available in PySindy (de Silva et al., 2020). Hence, all these methods crucially rely on regularly sampled and noise-free observations, whereas our approach can easily be extended to take those into account (see Appendix A).

- **Sindy** (Brunton et al., 2016): Sindy builds a (sparse) linear combination of a fixed set of (non-linear) basis functions. The resulting Lasso regression is efficient however complex expressions such as nested or parameterized functions can only be approximated but not easily be represented in their precise symbolic form as all non-linear expressions have to be explicitly added to the set of basis functions. We cross-validate Sindy over a fairly extensive hyperparameter grid of 800 different combinations for each individual trajectory.

- **GPL**[5] (genetic programming): GPL(earn) maintains a population of programs each representing a mathematical expression. The programs are mutated for several generations to heuristically optimize a user defined fitness function. While not originally developed for ODEs, we can apply GPLearn on our datasets by leveraging the finite difference approximation. We use a population

---

[5]gplearn.readthedocs.io/

Table 2: Comparing NSODE to the baselines on various benchmark datasets.

| Dataset | Metric | NSODE | Sindy | GPLearn | AIFeynman |
|---------|--------|-------|-------|---------|-----------|
| iv-163 | skel-recov | **37.4** | 3.7 | 2.5 | 14.1 |
| | $R^2 \geq 0.999$ | 24.5 | 31.9 | 3.7 | **49.7** |
| | allclose | 42.3 | 25.8 | 14.7 | **55.8** |
| | skel-recov & $R^2 \geq 0.999$ | **15.3** | 3.1 | 1.8 | 13.5 |
| | skel-recov & allclose | **15.3** | 3.1 | 1.8 | 13.5 |
| | runtime [s] | 5.4 | **0.4** | 29 +22 | 1203.6 |
| classic | skel-recov | 11.5 | 0 | 3.8 | **46.2** |
| | $R^2 \geq 0.999$ | 57.7 | 57.7 | 23.1 | **88.5** |
| | allclose | 80.8 | 57.7 | 30.8 | **88.5** |
| | skel-recov & $R^2 \geq 0.999$ | 0 | 0 | 7.7 | **46.2** |
| | skel-recov & allclose | 0 | 0 | 7.7 | **46.2** |
| | runtime [s] | 5.2 | **0.6** | 23 +22 | 1291.6 |
| textbook | skel-recov | 41.7 | 33.3 | 8.3 | **91.7** |
| | $R^2 \geq 0.999$ | 16.7 | 50 | 0.0 | **75** |
| | allclose | 25 | 58.3 | 8.3 | **75** |
| | skel-recov & $R^2 \geq 0.999$ | 33.3 | 41.7 | 0 | **66.7** |
| | skel-recov & allclose | 8.3 | 33.3 | 1.8 | **66.7** |
| | runtime [s] | 6 | **1** | 23 +22 | 1267.1 |

size of 1000 and report the best performance across all final programs. Compared to sindy, GPLearn is more expressive yet much slower to fit.

- **AIFeynman** (Udrescu & Tegmark, 2020; Udrescu et al., 2020): AIFeynman is a physics-inspired approach to symbolic regression that exploits the insight that many famous equations in natural sciences exhibit well-understood functional properties such as symmetries, compositionality, or smoothness. AIFeynman implements a neural network based heuristic search that tests for such properties in order to identify a symbolic expression that fits the data. For every test sample AIFeynman computes a pareto front of solutions that trade off complexity versus accuracy. We report the best performance across all functions on the pareto front. Notably, AIFeynman performs quite an exhaustive search procedure such that running it even on a single equation took on the order of tens of minutes.

### 4.3 RESULTS

**Model Performance.** Figure 2 shows NSODE's performance on our testset-iv, testset-constants, and testset-skeletons according to our numerical and symbolic metrics as well as combined skeleton recovery and allclose as we vary $k$ in the top-k considered candidates of the beam search. Investing sufficient test-time-compute (i.e., considering many candidates) continuously improves performance. While we capped $k$ at 1536 due to memory limitations, we did not observe a stagnation of the roughly logarithmic scaling of all performance metrics with $k$. This cannot be attributed to "exhaustion effects", where one may assume that all possible ODEs will eventually be among the candidates, because (a) the space of possible skeletons grows much faster than exponentially, and (b) the numerical metrics are extremely sensitive also to the predicted constant values in continuous domains. We verified this in an ablation experiment in Appendix F.

As one may expect, performance decreases as we move from only new initial conditions, to also sampling new constants, to finally sampling entirely unseen skeletons. On testset-iv with $k = 1536$ we achieve about 50% skeleton recovery and still successfully recover more than a third skeletons of testset-skeletons with similar numbers for allclose. The fact that the combined metric (symbolic + numerical) is only about half of that indicates that numerical and symbolic fit are indeed two separate measures, none of which need to imply the other. Hence, a thorough evaluation of both is crucial to understand model performance in symbolic regression tasks.

**Comparison to baselines.** In Table 2 we compare NSODE to all baselines (see Section 4.2) on all our metrics (see Section 3.3) on all test sets (see Section 4.1) using $k = 1536$ in our beam search. We also include the average wallclock runtime per expression for each of the datasets. Since GPLearn

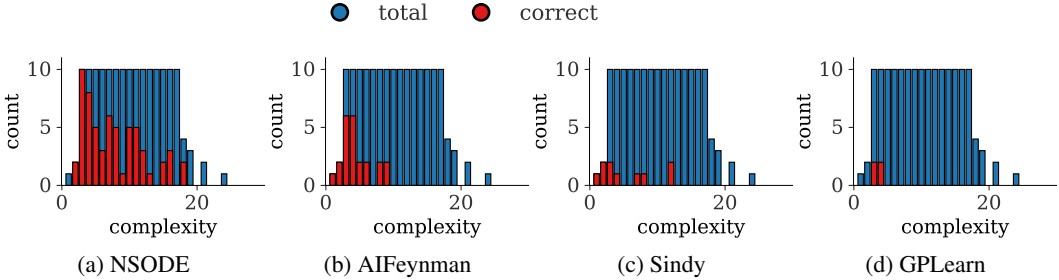

Figure 3: Correctly recovered skeletons by each method on testset-iv-163 per complexity. AIFeynman and Sindy are mostly able to recover some of the low complexity skeletons, while NSODE performs much better also on higher complexities. GPLearn fails to recover most skeletons.

often generates extremely long string expressions, it often takes SymPy up to half a minute to parse these expressions. We denote this extra time in gray and provide a more detailed version in Table 9.

First, we note that on our subsampled testset-iv-163, NSODE outperforms competing approaches in terms of skeleton recovery by a wide margin and also performs best in terms of joint skeleton recovery and numerical measures, which is a strong indication of actually having recovered the governing ODE accurately. By spending over 200x more time on its exhaustive heuristic search, AIFeynman manages to outperform NSODE in terms of numerical accuracy ($R^2$ and allclose), which again does not necessarily correspond to correct identification of the ODE structure. Figure 3 shows the number of skeletons recovered by each method given the complexity of equations, results for dataset-classic and dataset-textbook can be found in Appendix G. [6] While AIFeynman and Sindy recover some of the low complexity expressions, NSODE is the only method to also recover some of the more complex skeletons.

This also explains the strong performance on AIFeynman on the overwhelmingly simple expressions in the testset-classic and testset-textbook. The heuristics deployed by AIFeynman are tuned to incorporate properties of known equations in the sciences. In particular, the algorithm explicitly attempts to fit a polynomial to the data which immediately corresponds to the correct ground truth expression in $8/12 \approx 66.6\%$ equations in testset-textbook and $12/26 \approx 46.2\%$ equations in testset-textbook. Together with the fact that it is solving a somewhat simpler task (identifying functional relationships with manually provided derivatives from noise-free regularly sampled time-series), its strong performance on these particular test cases is to be expected. However, even on these simple examples AIFeynman takes over 200x longer than our method, which in turn clearly outperforms Sindy and GPLearn in terms of skeleton recovery.

## 5 CONCLUSION

We have developed a flexible and scalable method to infer ordinary differential equations $\dot{y} = f(y)$ from a single observed solution trajectory. Our method follows the successful paradigm of large-scale pretraining of attention-based sequence-to-sequence models on essentially unlimited amounts of simulated data, where the inputs are the observed solution $\{(t_i, y_i)\}_{i=1}^n$ and the output is a symbolic expression for $f$ as a string. Once trained, our method is orders of magnitude more efficient than similarly expressive existing symbolic regression techniques that require a separate optimization for each instance and achieves strong performance in terms of skeleton recovery especially for complex expressions on various benchmarks. Current limitations of our method mostly concern extensions to systems of high-order non-autonomous equations $f$, which we discuss in detail in Appendix A. While these extensions are possible in principle with few to no modifications of NSODE, it is an interesting direction for future to assess empirically how far this approach can be pushed. Besides the potential usefulness of our model in real-world scientific discovery and hypothesis generation, we also hope that our released dataset, code, and checkpoints will serve as a useful starting point for further research in this direction.

---

[6]We note that the maximum complexity is not determined by $K$, maximum number of internal nodes in the tree alone, since we also simplify all expressions using SymPy sometimes leading to higher complexities.

## 6 ETHICS STATEMENT

This paper addresses fundamental research and as such its implications are difficult to forsee. However, all of our experiments are based on non-personalized data so that we believe that datasets and model checkpoints can be publicly released without compromising private information.

## 7 REPRODUCIBILITY STATEMENT

To facilitate reproducibility we will release all datasets, source code to our model and baseline implementations as well as our final model checkpoint. We also report all hyperparameter choices for our model in Section 3.2 and Appendix B.3 as well as for the baseline methods Appendix E. Hyper-parameters for data generation are described in Section 3.1 and concrete choices are listed in Appendix B.2.

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

## A  POSSIBLE EXTENSIONS

While we have focused exclusively on the huge class of explicit scalar, autonomous, first-order ODEs, we believe that our approach can scale also to non-autonomous, implicit, higher-order systems of ODEs.

**Non-autonomous equations.** Since our model is provided with time $t$ as inputs, it is capable of learning functions $f(y, t)$ depending on $y$ and $t$ explicitly. Hence, extending our approach to non-autonomous ODEs is simply a matter of adding a symbol for $t$ in our data generation.

**Implicit equations.** A generalization from purely explicit ODEs to all ODEs of the form $f(\dot{y}, y) = 0$ are possible but rely on the availability of appropriate ODE solvers for implicit equations. Sampling $f$ of this general form using the sampling algorithm described in Section 3.1 is straightforward as we only need to include the additional variable $\dot{y}$ in the set of candidate symbols and its associated distribution $p_{\text{sym}}$. If we have an appropriate implicit numerical solver at hand, we can supply the numerical solution trajectory to NSODE as before and use $f$ for teacher forcing during training.

**Irregular samples and noise.** Due to the separate data generation and training phase, it is straight forward to train NSODE on corrupted input sequences, where we could for example add observation noise (need not be additive) to the $\{y_i\}_{i=1}^n$ or randomly drop some of the observations on the regular observation grid to simulate irregularly sampled observations. We expect these models to still perform well and in particular much better than the baselines, since their targets are approximated derivatives, which are highly sensitive to noise and irregular samples.

**Systems of equations.** For a system of $K$ first-order ODEs, the data generation requires two updates: first, we have to generate $K$ equations $(f_k)_{k \in \{1,...,K\}}$, one for each component. In addition, each of those functions can depend not only on $y$ (and $t$ in the non-autonomous case), but $K$ different components $\{y^{(k)}\}_{k \in \{1,...,K\}}$. This is easily achieved by allowing for $K$ tokens $y^{(k)}$ in the data generation. We can then simply augment the input sequence to the transformer to contain not only $(t, y)$ as for the scalar case, but $(t, y^{(1)}, y^{(2)}, \ldots, y^{(K)})$ for a system of $K$ variables. Finally, we have decide on a way to set the target for the transformer, i.e., how to represent the system of equations in symbolic form as *a single sequence*. A straight forward way to achieve this is to introduce a special delimitation character, e.g., `"|"` to separate the different components. Note that the order in which to predict the $K$ equations is dictated by the order in which they are stacked for the input sequence, hence this information is in principle available to the model. For example, in the case of a two-dimensional system, where $y = (y^{(1)}, y^{(2)}) \in \mathbb{R}^2$, we could have the model perform the following mapping

$$\{(t_i, y_i^{(1)}, y_i^{(2)})\}_{i=1}^n \to \text{"-1.4*y}^{(1)}\text{*y}^{(2)}\text{+cos(t) | sin(y}^{(1)}\text{+y}^{(2)}\text{)"},$$

where the separator | delimits the two components of $f(y, t)$ corresponding to $\dot{y}^{(1)}$ and $\dot{y}^{(2)}$.

Although the extension to systems is conceptually straightforward, we remark that the performance of the approach needs to be carefully evaluated in future work as systems of ODEs show a much larger variety of qualitative behaviors - including chaos. One of the defining characteristics of chaos is the sensitivity of solution trajectories to initial values (Strogatz, 2018), that is, even similar initial values

result in drastically different trajectories. Chaotic behavior may pose a fundamental limitation for NSODE, which relies on supervised optimization on a representative training dataset, a challenging requirement in case of chaotic systems.

**Higher-order equations.** Finally, it is well known that any higher order (system of) ODEs can be reduced to a first-order system of ODEs. Specifically, a $d$-th order system of $K$ variables can be reduced to an equivalent first-order system of $d \cdot K$ variables. Hence, one can handle higher-order systems analogously as before with multiple separator tokens. One obstacle in this case is that when only observations of $y(t)$ are given, one first needs to obtain observed derivatives to reduce a higher-order system to a first-order system. These would in turn have to be estimated from data, which suffers from the same challenges we have mentioned previously (instability under noise and irregular samples).

Finally, when we want to have a single model deal with higher-order equations of unknown order, or systems with differing numbers of variables, it remains on open question how to have the model automatically adjust to the potentially vastly differing input dimensions or how to automatically detect the order of an ODE.

# B    IMPLEMENTATION DETAILS

## B.1    RULES TO RESAMPLE CONSTANTS

As described in Section 3.1, we generate ODEs as unary-binary trees, convert them into infix notation and parse them into a canonical form using `sympy`. From each skeleton we then create up to 25 ODEs by sampling different values for the constants. When resampling constants we want to ensure that we do not accidentally modify the skeleton as this would additionally burden our model with resolving potential ambiguities in the grammar of ODE expressions. Furthermore, we do not want to reintroduce duplicate samples on the skeleton level after carefully filtering them out previously. We therefore introduce the following sampling rules for constants:

1. Do not sample constants of value 0.

2. When the original constant in the skeleton is negative, sample a negative constant, otherwise sample a positive constant.

3. Avoid base of 1 in power operations as $1^x = 1$.

4. Avoid exponent of 1 and -1 in power operations as $x^1 = x$ and $x^{-1} = 1/x$.

5. Avoid coefficients of value 1 and -1 as $1 \cdot x = x$ and $-1 \cdot x = -x$

6. Avoid divisions by 1 and -1 as $x/1 = x$ and $x/-1 = -x$

## B.2    DATA GENERATION

As discussed in the main text, the choices of the maximum number of internal nodes per tree $K$, the choice and distribution over $N_{\text{bin}}$ binary operators, the choice and distribution over $N_{\text{una}}$ unary operators, the probability with which to decorate a leaf with a symbol $p_{\text{sym}}$ (versus a constant with $1 - p_{\text{sym}}$), and the distribution $p_{\text{c}}$ over constants uniquely determine the training distribution over ODEs $f$. These choices can be viewed as flexible and semantically interpretable tuning knobs to choose a prior over ODEs. For example, it may be known in a given context, that the system follows a "simple" law (small $K$) and does not contain exponential rates of change (do not include $\exp$ in the unary operators), and so on. The choice of the maximum number of operators per tree, how to sample the operators, and how to fill in the leaf nodes define the training distribution, providing us with flexible and semantically meaningful tuning knobs to choose a prior over ODE systems for our model. We summarize our choices in Tables 3 to 5, where $\mathcal{U}$ denotes the uniform distribution. Whenever a leaf node is decorated with a constant, the distribution over constants is determined by first determining whether to use an integer or a real value with equal probablity. In case of an integer, we sample it from $p_{\text{int}}$, and in case of a real-valued constant we sample it from $p_{\text{real}}$ shown in Table 3. Finally, when it comes to the numerical solutions of the sampled ODEs, we fixed the parameters in Table 6 for our experiments.

We highlight that there is no such thing as "a natural distribution over equations" when it comes to ODEs. Hence, ad-hoc choices have to be made in one way or another. However, it is important to note that neither our chosen range of integers nor the range of real values for constants are in any way restrictive as they can be achieved by appropriate rescaling. In particular, the model itself represents these constant values merely be non-numeric tokens and interpolates between those anchor tokens (our two-hot encoding) to represent continuous values. Hence, the model is entirely agnostic to the actual numerical range spanned by these fixed grid tokens, but the relative accuracy in recovering interpolated values will be constant and thus scale with the absolute chosen range. Therefore, scaling $p_{\text{int}}$ and $p_{\text{real}}$ by essentially any power of 10 does not affect our findings. Similarly, the chosen range of initial values $(y_0^{\min}, y_0^{\max})$ is non-restrictive as one could simply scale each observed trajectory to have its starting value lie within this range.

### B.3 Model

For our Transformer model we choose the implementation of BigBird (Zaheer et al., 2020) available in HuggingFace. The model is trained on 4 Nvidia A100 GPUs for 18 epochs after which we evaluate the best model based on the validation loss. We choose a batchsize of 600 samples and use a linear learning rate warm-up over 10,000 optimization step after which we keep the learning rate constant at $10^{-4}$. For the fixed tokens that are used to decode constants, we choose an equidistant grid $-10 = x_1 < x_2 < \ldots < x_m = 10$ with $m = 21$. This worked well empirically and using fewer or more tokens did not seem to improve model performance substantially. Finally, we use $N_{\text{eval}} = 100$ for the evaluation of our numerical metrics. We did not optimize architectural choices or hyperparameters for maximum performance and use the same choices in all experiments.

While not relevant for our dataset as we check for convergence of the ODE solvers, we remark that the input-encoding via IEEE-754 binary representations also graciously represents special values such as `nan` or `inf` without causing errors. Those are thus valid inputs that may still provide useful training signal, e.g., "the solution of the ODE of interest goes to `inf` quickly".

## C Textbook equations dataset

Table 7 list the equations we collected from wikipedia, textbooks and lecture notes together with the initial values that we solved them for. We can also see that almost all of these equations simplify to low-order polynomials.

## D Dataset statistics

We provide an overview over the complexity distribution and the absolute frequency of all operators (after simplification) for all datasets in Figure 4. We can see that our self-generated dataset covers by far the larges complexity whereas both complexities and operator diversity are much lower for equations in the classic and textbook ODEs.

## E Baselines

We here describe more detail on the optimization of the baseline comparison models.

**Sindy.** We use the implementation available in PySindy (de Silva et al., 2020) and instantiate the basis functions with polynomials up to degree 10 as well as all unary operators listed in Table 5. When fitting sindy to data we often encountered numerical issues especially when using high-degree polynomial or the exponential function. To attenuate such issues we set the highest degree of the

Table 3: Parameter settings for the data generation.

| parameter | $K$ | $N_{\text{bin}}$ | $N_{\text{una}}$ | $p_{\text{sym}}$ | $p_{\text{int}}$ | $p_{\text{real}}$ |
|---|---|---|---|---|---|---|
| value | 5 | 5 | 5 | 0.5 | $\mathcal{U}(\{-10, \ldots, 10\} \setminus \{0\})$ | $\mathcal{U}((-10, 10))$ |

Table 4: Binary operators with their relative sampling frequencies

| operator | $+$ | $-$ | $\cdot$ | $\div$ | pow |
|---|---|---|---|---|---|
| probability | 0.2 | 0.2 | 0.2 | 0.2 | 0.2 |

Table 5: Unary operators with their relative sampling frequencies.

| operator | sin | cos | exp | $\sqrt{\ }$ | log |
|---|---|---|---|---|---|
| probability | 0.2 | 0.2 | 0.2 | 0.2 | 0.2 |

Table 6: Parameters for numerical solutions of sampled ODEs.

| parameter | $N_{\text{const}}$ | $N_{\text{iv}}$ | $T$ | $N_{\text{grid}}$ | $(y_0^{\min}, y_0^{\max})$ |
|---|---|---|---|---|---|
| value | 25 | 25 | 4 | 1024 | $(-5, 5)$ |

Table 7: Equations of the textbook dataset.

| Name | Equation $f(x)$ | simplified | $y_0$ |
|---|---|---|---|
| autonomous Riccati | $0.6 \cdot y^2 + 2 \cdot y + 0.1$ | $0.6 \cdot y^2 + 2 \cdot y + 0.1$ | $-0.2$ |
| autonomous Stuart-Landau | $-2.2/2 \cdot y^3 + 1.31 \cdot y$ | $-1.1 \cdot y^3 + 1.31 \cdot y$ | 0.1 |
| autonomous Bernoulli | $-1.3 \cdot y + 2.1 \cdot y^{2.2}$ | $-1.3 \cdot y + 2.1 \cdot y^{2.2}$ | 0.6 |
| compound interest | $0.1 \cdot y$ | $0.1 \cdot y$ | 9 |
| Newton's law of cooling | $-0.1 \cdot (y - 3)$ | $0.3 - 0.1 \cdot y$ | 9 |
| Logistic equation | $0.23 \cdot y \cdot (1 - y)$ | $0.23 \cdot (y - y^2)$ | 9 |
| Logistic equation with harvesting | $0.23 \cdot y \cdot (1 - 0.33 \cdot y) - 0.5$ | $0.23 \cdot y - 0.76 \cdot y^2 - 0.5$ | 9 |
| Logistic equation with harvesting 2 | $2 \cdot y \cdot (1 - y/3) - 0.5$ | $2 \cdot y - 0.66 \cdot y^2 - 0.5$ | 0.7 |
| Solow-Swan | $y^0.5 \cdot (0.9 \cdot 8 - (3 + 2.5) \cdot y^{1-0.5})$ | $7.2 \cdot y^{0.5} - 5.5 \cdot y$ | 0.1 |
| Tank draining | $-\sqrt{2 \cdot 9.81} \cdot (2/9)^2 \cdot \sqrt{y}$ | $-0.21 \cdot y^{0.5}$ | 1 |
| Draining water through a funnel | $-(0.5^2/4) \cdot \sqrt{2 \cdot 9.81} \cdot (\sin 1 / \cos 1)^2 \cdot y^{-1.5}$ | $-0.67/y^{1.5}$ | 3 |
| velocity of a body thrown vertically upwards | $-9.81 - 0.9 \cdot y/8.2$ | $-0.10 \cdot y - 9.81$ | 0.1 |

polynomials per sample to the highest degree present in the ground truth. Secondly, when numerical issues are caused by a particular basis function, we discard this basis function for the current sample and restart the fitting process. We run a separate full grid search for every ODE over the following hyper-parameters and respective values (these all include the default values):

- optimizer-threshold (`np.logspace(-5, 0, 10)`): Minimum magnitude for a coefficient in the weight vector to not be zeroed out.
- optimizer-alpha ($[0.001, 0.0025, 0.005, 0.01, 0.025, 0.05, 0.1, 0.2]$): L2 regularizer on parameters.
- finite differences order ($[2, 3, 5, 7, 9]$): Order of finite difference approximation.
- maximum number of optimization iterations ($[20, 100]$): Maximum number of optimization steps.

For every ODE, sindy is fit using solution trajectory in the initial interval $[0, 2]$ and validated on the interval $(2, 4]$. The grid search thus results in a ranking of models with different hyper-parameter configurations. Instead of evaluating only the performance of the best model, we report top-k performance across the ranked hyper parameter configurations. Sindy is computationally highly efficient yet complex expressions such as nested or parameterized functions can only be approximated (e.g. by a polynomial basis) but not represented in their exact symbolic form.

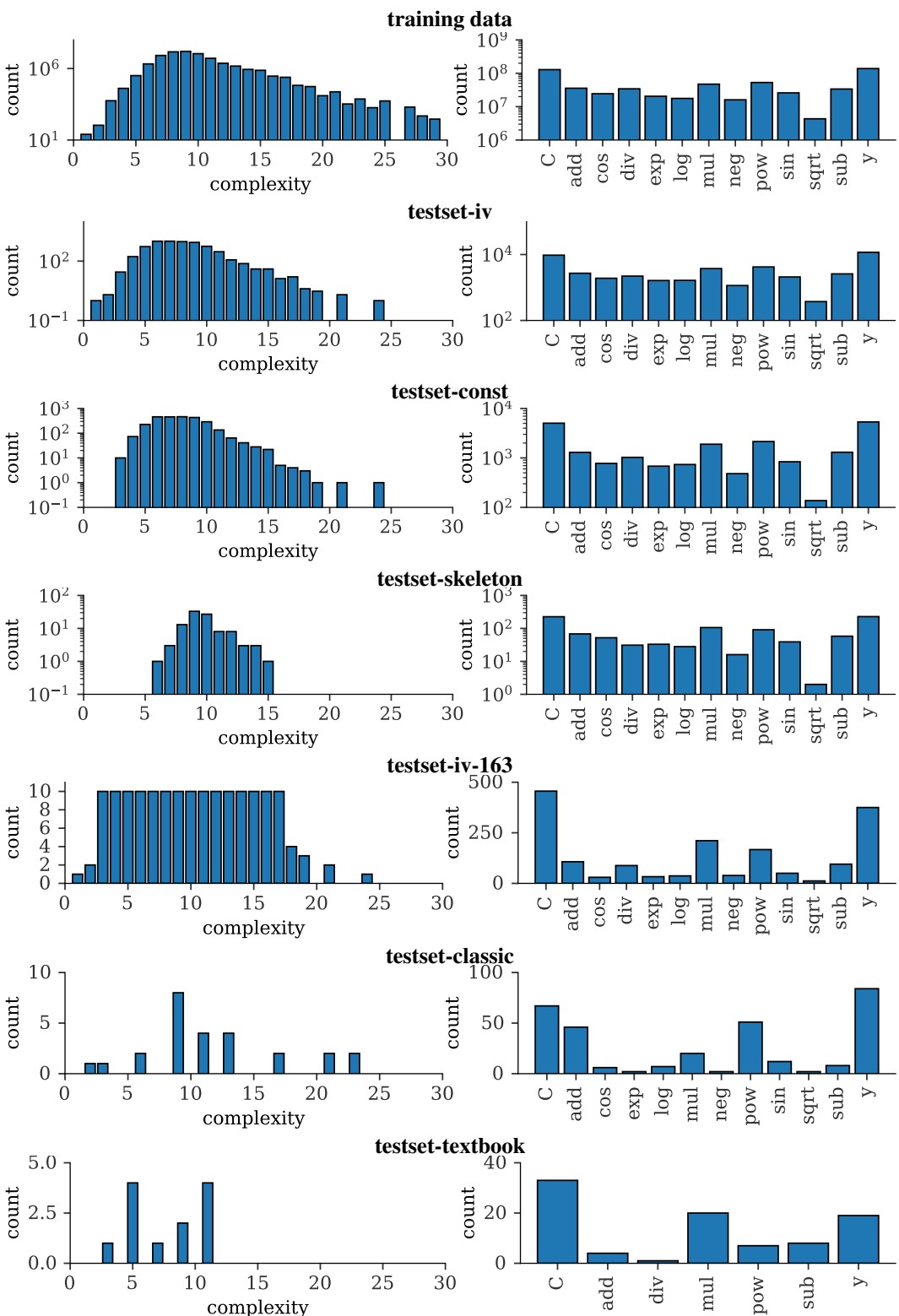

Figure 4: Distribution of complexity and operators for all datasets.

**GPLearn.** We instantiate GPLearn with a constant range of $(-10, 10)$ and all binary operators listed in Table 4 and all unary operators listed Table 5 except for the exponential function which caused numerical issues. We keep the default hyper-parameters but run a grid search across the parsimony

coefficient ($\{0.0005, 0.001, 0.01, 0.05, 0.1, 0.2, 0.5, "auto"\}$) which trades of fitness versus program length. We choose $R^2$ as fitness function.

**AIFeynman.** We use the AIFeynman implementation from https://github.com/SJ001/AI-Feynman and run the algorithm with the following (default) hyper-parameters:

- Brute force try time: 60 seconds
- Number of epochs for the training : 500
- Operator set: 14
- Maximum degree of polynomial tried by the polynomial fit routine: 4

# F    ABLATION

The experimental results presented in Section 4.3 show that the performance measured by all metrics improves with increases in hyperparameter $k$ of the top-k sampling algorithm. While this result is intuitive, it also raises the question whether the model simply generates a diverse set of random guesses that cover the full diversity of the testset. We investigated this question via the following ablation experiment: If we assume that the model only generates a diverse set of random guesses, then we should be able to permute predictions across samples without loss in performance, e.g.

$$(f_1, \{\hat{f}_{1,1}, ..., \hat{f}_{1,K}\}) \xrightarrow{\text{permute}} (f_1, \{\hat{f}_{3,1}, ..., \hat{f}_{3,K}\})$$

$$(f_2, \{\hat{f}_{2,1}, ..., \hat{f}_{2,K}\}) \longrightarrow (f_2, \{\hat{f}_{4,1}, ..., \hat{f}_{4,K}\})$$

$$\vdots \qquad\qquad\qquad \vdots$$

$$(f_{5793}, \{\hat{f}_{5793,1}, ..., \hat{f}_{5793,K}\}) \longrightarrow (f_{5793}, \{\hat{f}_{2,1}, ..., \hat{f}_{2,K}\})$$

where $f_n$ is the ground truth symbolic expression for sample $n$ and $\{\hat{f}_{n,1}, ..., \hat{f}_{n,K}\}$ correspond to the generated predictions for sample $n$ with top-k=$K$.

We implemented this experiment using the results obtained on testset-iv for $k = 1536$. Contrary to our assumption Table 8 shows that the resulting performance drops substantially in comparison to the original performance on all metrics. We can therefore conclude that our model is not simply generating diverse random guesses but actually learned a systematic mapping from input observations to symbolic expressions. Similarly, we can conclude that samples in the dataset are sufficiently distinct from each other, i.e. that the data generation process presented in Section 3.1 resulted a diverse dataset.

Table 8: Comparison between original performance and performance after permuting predictions across samples on testset-iv with k = 1536. Results are given as percentages.

| Metric | Original | Permuted |
|---|---|---|
| skel-recov | 52.0 | 0.8 |
| $R^2 \geq 0.999$ | 28.6 | 0.4 |
| allclose | 50.6 | 6.5 |
| skel-recov & $R^2 \geq 0.999$ | 17.0 | 0 |
| skel-recov & allclose | 22.6 | 0.1 |

# G    DETAILED RESULTS

We provide a comprehensive summary of performances of all models on all datasets in Table 9. Figures 5 and 6 further show again the number of correctly recovered skeletons by each method per complexity.

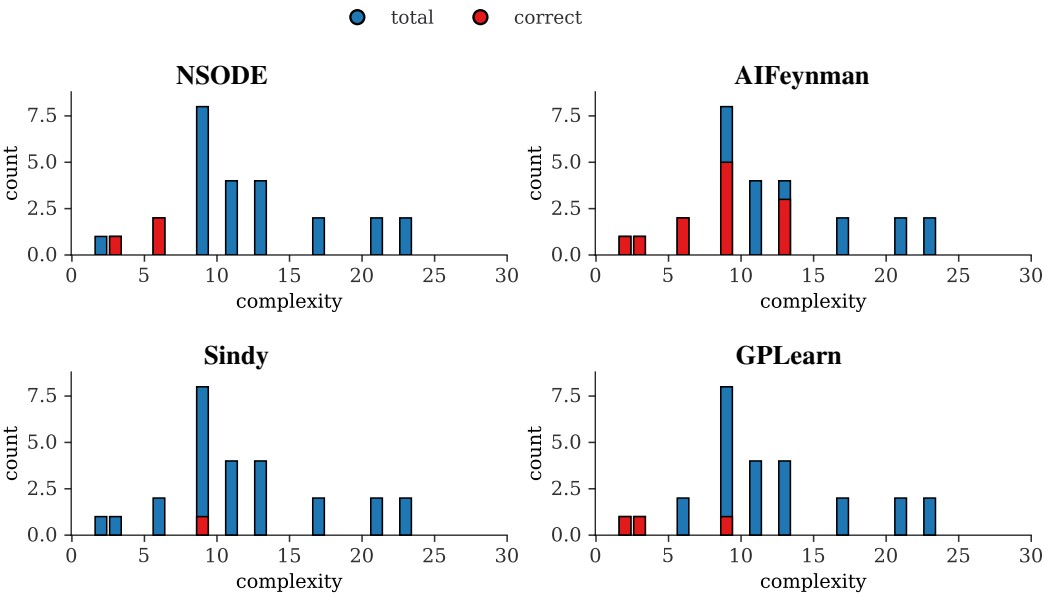

Figure 5: Correctly recovered skeletons by each method on the classic benchmark dataset per complexity.

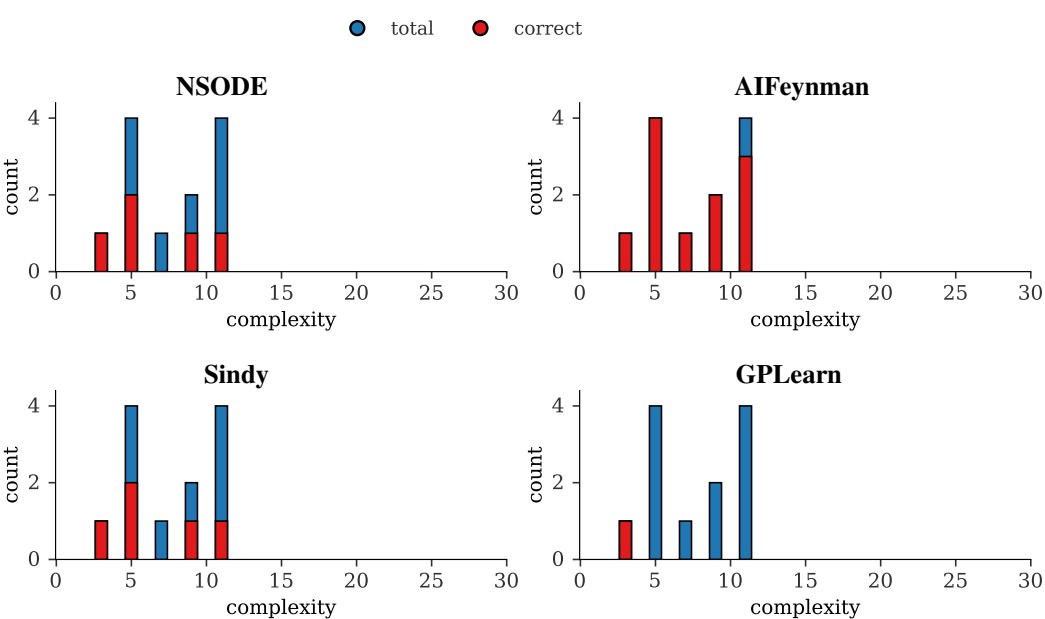

Figure 6: Correctly recovered skeletons by each method on the textbook equation dataset per complexity.

Table 9: Detailed performance results for all methods on all (applicable) datasets.

| Dataset | Metric | NSODE | Sindy | GPLearn | AIFeynman |
|---|---|---|---|---|---|
| | skel-recov | 52.0 | - | - | - |
| | $R^2 \geq 0.999$ | 28.6 | - | - | - |
| testset-iv | allclose | 50.6 | - | - | - |
| | skel-recov & $R^2 \geq 0.999$ | 17 | - | - | - |
| | skel-recov & allclose | 22.6 | - | - | - |
| | runtime [s] | 5.3 | - | - | - |
| | skel-recov | 45.6 | - | - | - |
| | $R^2 \geq 0.999$ | 21.7 | - | - | - |
| testset-constant | allclose | 44.7 | - | - | - |
| | skel-recov & $R^2 \geq 0.999$ | 9.8 | - | - | - |
| | skel-recov & allclose | 16.1 | - | - | - |
| | runtime [s] | 5.3 | - | - | - |
| | skel-recov | 19 | - | - | - |
| | $R^2 \geq 0.999$ | 12 | - | - | - |
| testset-skel | allclose | 33 | - | - | - |
| | skel-recov & $R^2 \geq 0.999$ | 1 | - | - | - |
| | skel-recov & allclose | 2 | - | - | - |
| | runtime [s] | 5.3 | - | - | - |
| | skel-recov | **37.4** | 3.7 | 2.5 | 14.1 |
| | $R^2 \geq 0.999$ | 24.5 | 31.9 | 3.7 | **49.7** |
| testset-iv-163 | allclose | 42.3 | 25.8 | 14.7 | **55.8** |
| | skel-recov & $R^2 \geq 0.999$ | **15.3** | 3.1 | 1.8 | 13.5 |
| | skel-recov & allclose | **15.3** | 3.1 | 1.8 | 13.5 |
| | runtime [s] | 5.4 | **0.4** | 29 +22 | 1203.6 |
| | skel-recov | 11.5 | 0 | 3.8 | **46.2** |
| | $R^2 \geq 0.999$ | 57.7 | 57.7 | 23.1 | **88.5** |
| classic | allclose | 80.8 | 57.7 | 30.8 | **88.5** |
| | skel-recov & $R^2 \geq 0.999$ | 0 | 0 | 7.7 | **46.2** |
| | skel-recov & allclose | 0 | 0 | 7.7 | **46.2** |
| | runtime [s] | 5.2 | **0.6** | 23 +22 | 1291.6 |
| | skel-recov | 41.7 | 33.3 | 8.3 | **91.7** |
| | $R^2 \geq 0.999$ | 16.7 | 50 | 0.0 | **75** |
| textbook | allclose | 25 | 58.3 | 8.3 | **75** |
| | skel-recov & $R^2 \geq 0.999$ | 33.3 | 41.7 | 0 | **66.7** |
| | skel-recov & allclose | 8.3 | 33.3 | 1.8 | **66.7** |
| | runtime [s] | 6 | **1** | 23 +22 | 1267.1 |

