# OpenReview forum: "Efficient Discovery of Dynamical Laws in Symbolic Form"
_ICLR.cc/2023/Conference — Submitted to ICLR 2023_

### Official Review · Reviewer_LsD3 · 2022-10-19

**Confidence:** 5
**Correctness:** 4
**Technical Novelty And Significance:** 3
**Empirical Novelty And Significance:** 4
**Recommendation:** 8

**Clarity, Quality, Novelty And Reproducibility:**

**Clarity** - The paper is extremely clear, and very well written. The supplementary material is helpful.

**Quality** - The architecture and data generation techniques are solid. The experiments and comparisons with baselines are sound.

**Novelty** - This is, to my knowledge, the first case of symbolic regression recovering ODE. It is also one of the first papers proposing an end-to-end transformer-based solution without fine-tuning of the constants. That architecture and the data generation techniques are based on previous work, but the two-hot encoding of constants is an interesting innovation.

**Reproducibility** - Details are provided, both on the architecture and the data generation process, that allow to reproduce the results. The authors agree to make the model and dataset public. The model and datasets will certainly be useful for other research teams.

**Strength And Weaknesses:**

**Strengths**

The paper adresses a hard and important problem of science: given numerical measurements over time, discover the underlying dynamics.

The paper is clearly written. The architecture and data generation techniques are clearly described and sound.

The experiments are well-thought, and support the claims. The comparisons with baselines are adequate, and I am very happy to see a negative result here (i.e. AIFeynman beating the model), together with a likely explanation of why this is so.

Overall, a very solid paper, on an important subject.


**Weaknesses**

There are no major weaknesses or flaws in the paper. I have a number of questions and suggestions for the authors.

* **Limitations of the class of ODE**: at present, the only ODE considered have the form y'=f(y). A more general class of autonomous ODE would have the form f(y',y)=0, which could be generated with the same procedure, but will feature more difficult cases (i.e. singularities). The authors should mention this limitation, and possibly discuss it in appendix A.
* **Simplifying training functions**: all the functions in the training and test set are simplified using Sympy. D'Ascoli et al. (https://proceedings.mlr.press/v162/d-ascoli22a.html, appendix C) observe that this is not necessary. On the other hand, it slows the generation procedure, and might make the model less robust since it reduce the diversity of the expressions used for training. Have the authors tried to train the model without simplification?
* **Model ablation**: at present, the model uses a symmetric architecture, with 6 layers in the encoder and decoder. Some authors (e.g. https://arxiv.org/abs/2112.01898) have reported good results with asymmetric architectures. In this case, it seems that the decoder task is much harder, so maybe a shallow encoder and a deeper decoder would make sense. Kamienny uses a 16 layer decoder, and a 4 layer encoder. Have the authors experimented with such architectures? Also the models also use a fairly low dimension (512) and a large number of heads (16). Are there ablation studies justifying these choices?
* **Scaling**: the models are relatively small (46 M parameters, 6 layers and 512 dimensions). On simpler problems, d'Ascoli uses 8 layers, and Kamienny 83M parameters. Experimenting with slightly larger models might improve performance, especially in the "out of domain" case of new skeletons (testset-skeleton).
* **test-skeleton**: this test set is slightly smaller than the others. However, it provides the most practical assessment of model accuracy, since it measures model performance when the solution skeleton is not in the training set, a very likely case in practical situations. Could the authors provide a larger set when they make their dataset available?
* **beam search**: experimental results suggest that very large beam sizes are needed for the model to perform correctly. D'Ascoli (https://proceedings.mlr.press/v162/d-ascoli22a.html, section 2.3) proposes a technique for ranking the hypotheses in the beam, and other techniques such as MCTS have been discussed as possible improvements over top-k perplexity beam search. Have the authors considered such approaches?
* **benchmarks and out-of-domain generalization**: as the authors note, the better performance of AI-Feynman on benchmark sets might be explained by out-of-distribution generalization. These test sets feature much simpler functions that the ones seen at training. Have the authors considered retraining their model on a dataset of simpler functions (reducing the number of operators, and maybe changing the probabilities, in the algorithm they use), to provide for a better comparison (and support their interpretation of the benchmark results)?

**Summary Of The Paper:**

The authors propose a transformer-based model to recover first order ordinary differential equations (ODE) in symbolic form, from points sampled from their trajectories at N regular time steps. In the paper, the trajectories are supposed to be noiseless, save perhaps for approximation error when solving the ODE, or rounding error of the values sampled on the trajectory. The ODE considered are of the form y' = f(y).

The model is a sequence to sequence 6 layer transformer (using HuggingFace implementation of BigBird as the base layer), with 512 dimensions and 16 heads (47M parameters overall). Training data are generated by sampling random functions f, using the "tree algorithm" introduced by Lample & Charton (2019), symplifying f with Sympy, solving the corresponding equation y' = f(y) with a numerical solver (LSODA), and sampling the corresponding trajectories for a number of initial values. The model loss is the cross-entropy of the sequence corresponding to f (after preorder enumeration of the corresponding tree). Beam search is used for generation.

The authors evaluate their model on three test sets:
* solutions from the training set, but different initial values
* solution skeletons from the training set, with different initial values, and constants
* skeletons not found in the training set
They show that, with a large enough beam size, the model can reach significant accuracy (around 40%), both when numerically approximating f and recovering its skeleton.

They also provide (smaller) benchmark samples, and compare their model with three popular implementations : Sindy, GPLearn and AIFeynman. Their model outperforms Sindy and GPLearn, but AIFeynman, while over 200 times slower, is much more accurate. The authors suggest that this is due to differences between the training distribution and the benchmark samples (an out-of-distribution generalization problem).

**Summary Of The Review:**

This is a very strong paper, on an important subject. It is clearly written, the experimental setting is very good, and the results are compelling.

I recommend acceptance, and would be happy to increase my rating if the questions asked in the "weaknesses" sections are adressed during the rebuttal phase.

---

> ### Author Response · Authors · 2022-11-19
> **Thank you for the positive review, we are happy that you like our work. We will address your questions point-by-point below**
>
> **1. Limitations of the class of ODE**
> Thank you for the suggestion, we have added a paragraph to discuss ODEs of the form f(y, y’) = 0 in appendix A.
>
>
> **2. Simplifying training functions**
> We simplify expressions in order to be able to discard mathematical equivalent expressions, i.e., expressions that differ while being mathematically equivalent. The tree sampling algorithm presented by Lample & Charton (2019) samples any tree configuration (under the provided hyper-parameters) with equal probability. However, the number of trees that correspond to the same mathematical expression differs across expressions. For example, assume that we sample trees with maximally 2 operators. Then there is only a single tree that corresponds to the expression „exp(sin(y))“ whereas there are many trees that correspond to the expression $y$, e.g. $y + y - y$, $y - y + y$, $y \cdot y / y$ , … (for simplicity we write them here in infix notation). Simplification allows us to remove these duplicates and thus to ensure a diverse dataset and performance evaluation that is not biased towards particular overrepresented expressions. While training a model on non-simplified data is possible, performance evaluation on a non-simplified dataset may be misleading due to imbalanced expression distributions. We therefore only used simplified expressions.
>
>
> **3. Model ablation**
> When studying the literature we found approaches where the decoder has more layers than the encoder (Kamienny et al. (2022)) as well as approaches where the number of layers in encoder and decoder is the same (Vastl et al. (2022), Biggio et al. (2021), D’Ascoli et al. (2022), Lample & Charton (2019). An ablation study on the effect of layers (or even generally the architectural components) would therefore certainly be useful. However, such an ablation is beyond our computational resources. For this reason we had to make ad-hoc architectural choices and followed Biggio et al. (2021) while increasing the number of heads. We have added a sentence in appendix B.3 to clarify this. It appears highly plausible but remains speculative that better performance may be achievable with a different architecture and/or more computational resources.
>
>
> **4. Scaling**
> Indeed, based on the literature, scaling up the architecture indeed seems to be a plausible and (conceptually) simple way to improve model performance. Our choices were mainly informed by our computational budget which did not allow us to further scale up the model.
>
>
> **5. Test-skeleton**
> We will provide a larger dataset. In addition we are also planning to publish the code for data generation to enable anyone interested to generate additional data.
>
>
> **6. Beam search**
> Thank you for bringing this to our attention. If we understand correctly, the procedure used by d’Ascoli et al. (2022) ranks generated symbolic expressions based on how closely they approximate the available observational data. Such a procedure is in principle also applicable to our model, although it would involve an additional integration step as we predict the ODE f with $\dot{y}(t) = f(y(t))$ whereas the observational data correspond to samples from the solution trajectory y(t). In the current paper, we chose top-k sampling for its simplicity, however, this choice can easily be replaced by other sampling algorithms, as mentioned in section 3.3 (first paragraph).
>
>
> **7. Benchmarks and out-of-domain generalization**
> Thank you for the suggestion. Indeed, retraining on e.g. only polynomials (or polynomials with fractional exponents) seems like a straightforward idea to improve performance of the proposed model on the textbook dataset. On the other hand, we would be tuning the model based on the testset, i.e. on information that is generally not available at inference time.

---

### Official Review · Reviewer_UR3m · 2022-10-21

**Confidence:** 4
**Correctness:** 3
**Technical Novelty And Significance:** 2
**Empirical Novelty And Significance:** 2
**Recommendation:** 3

**Clarity, Quality, Novelty And Reproducibility:**

I’m a bit undecided about this paper. On the one hand, it might be an interesting step toward deducing a symbolic representation of time series data in terms of governing equations. On the other hand, for its purported application domain, the (natural) sciences, I would say it’s hardly useful, and way more powerful approaches exist.

In more detail:

1) The approach is only developed for scalar, autonomous, noise-free, fully observed ODEs (although the authors allude to possible extensions in Appx. A, but my feeling is there is actually a long way to go). In most modern scientific applications, on the contrary, we deal with high-dimensional, noisy, partially observed, potentially non-stationary and non-autonomous systems. Autonomous scalar (first-order) ODEs cannot even produce oscillations, a bare minimum for many scientific applications. Especially for chaotic systems the type of symbolic inference proposed by the authors I think will become extremely challenging (if not impossible; for instance, accurate estimation of numerical constants will be become super-important). There is a huge body of work meanwhile in scientific machine learning (hardly reviewed here) which deals with inferring the governing equations from time series data for these much more complex and real-world situations. Besides the older Brunton paper, here are a few more pointers:
[1] https://www.nature.com/articles/s41598-022-13644-w
[2] https://arxiv.org/abs/2106.06898
[3] https://arxiv.org/abs/2207.02542
[4] https://arxiv.org/abs/2006.13431
[5] https://arxiv.org/pdf/2202.07022
[6] https://arxiv.org/abs/1712.09707
[7] https://openreview.net/pdf?id=aUX5Plaq7Oy
[8] https://arxiv.org/pdf/2201.05136.pdf
[9] https://arxiv.org/abs/2110.05266
[10] https://arxiv.org/abs/2207.00521

2) For any real-world situation, I’m not even sure the authors work with sensible optimization targets: In their artificial database, they have exact ground truth symbolic expressions. But in science of course one doesn’t have that, and there are usually several equivalent ways to formally describe “laws of nature” (i.e., nature does not assign one specific symbolic expression to any set of observations). It is, in my opinion, therefore not very relevant to recover the exact symbolic representation of some ground truth data for which such a symbolic expression exists. Rather, it is way more important to correctly recover the dynamical behavior supporting the system, i.e. the vector field and various of its topological and geometrical properties (see [3], [9], [10], and common textbooks like Strogatz). Yet, the authors, on the contrary, mostly emphasize the symbolic agreement. In addition, I didn’t find the % recovery of symbolic form even under quite ideal conditions (iv-163 with large k) overly impressive for any of the models.

3) Of course, one main point of the present paper is that many systems that have been used for extracting dynamical equations from data are not easily interpretable (e.g. NODE), and this is a valid point. I am therefore not contesting that, in principle, it makes sense to think along the lines followed by the authors (I just think the objectives need to be different for this to be scientifically useful). There are, however, also other approaches which are much more powerful (in the sense that they can deal with high-dim, noisy, partial etc systems) yet yield an interpretable form. Here, by ‘interpretable’ I mean how easily the recovered system can be analyzed and related to the data (which I think is more important from a scientific angle), not necessarily whether a specific symbolic form matches that of ground truth data (and given that we don’t have this for natural systems anyway). SINDy and its successors ([6], [8]) fall into this class: The statement that SINDy is not very expressive is wrong in my mind, since it can function as a universal approximator if equipped with the right basis (e.g. Stone-Weierstrass). Yet it yields an interpretable, sparse form. Other interpretable, in the sense above, systems for recovering dynamical laws have been based on piecewise-linear approximations [3].


**Strength And Weaknesses:**

Strengths:
- yields a human-interpretable explicit symbolic form for the governing equation generating a given time series

Weaknesses:
- puts too much emphasis on an exact symbolic form, rather than on dynamically important properties
- works only for very simple situations (scalar, noise-free, autonomous ...), unlike other methods developed in the dynamical systems field


**Summary Of The Paper:**

In this paper a transformer is trained to infer from a set of scalar time series observations the dynamical law behind these observations, in symbolic form. A large data set of equations and trajectories generated by these is generated, from which various test set scenarios are created. Performance is evaluated both in terms of the recovery of the “correct” symbolic expression, as well as numerically in terms of match between local time derivatives. The model is compared to various other approaches, and in general performs favorably, although it does not always outperform other models for reasons discussed.

**Summary Of The Review:**

The authors may follow a useful, complementary direction, different from that most commonly engaged in scientific ML. However, I believe it is partly based on the wrong objectives if application in science is aimed for. It also falls far behind the methods based on universal approximation in expressivity, power, and scalability. Finally, Fig. 3 is not overly impressive (for any of the models tested), and the drop-off in success with model complexity hints at fundamental limits for this class of models regarding scalability to more realistic and higher-dim scenarios.

---

> ### Author Response · Authors · 2022-11-19
> **Thank you for the thorough review and the detailed feedback. We are happy to hear that you find our work interesting and will address your concerns point-by-point below.**
>
> **1. Limitations to scalar, autonomous, noise-free, fully-observed ODEs**
>
> As described in the problem statement (section 3), we intentionally limited the development and evaluation of the proposed model in this paper to scalar, autonomous ODEs with noise-free and fully-observed data. As the performance evaluations of all models suggest, even under these settings inferring a symbolic function that faithfully captures the dynamics is a challenging problem. We agree that extensions beyond these assumptions are certainly interesting and relevant to make the model applicable to the scientific problem domain. We describe conceptually straightforward extensions to non-autonomous ODEs, systems of ODEs as well as noisy and missing data in appendix A. Their implementation and evaluation was beyond the scope for this paper.
>
> Thank you for your comment on chaotic systems! We agree that chaos may pose a fundamental challenge when extending our approach to systems. We have added a paragraph describing this limitation in appendix A. We are also very grateful for the many pointers to related work, especially for relevant papers that do not directly originate in the symbolic regression literature which was our starting point. We revised section 2 (Background and related work) to account for these and to clarify the advantages and disadvantages of different modeling approaches.
>
>
> **2. Evaluation on symbolic expressions**
>
> Thank you for this detailed comment! We will break down our response into multiple parts to address all aspects that you raised.
>
> Firstly, concerning ‘I’m not even sure the authors work with sensible optimization targets: In their artificial database, they have exact ground truth symbolic expressions. But in science of course one doesn’t have that [...]’ we would like to clarify that optimization targets are only required for training. Our proposed data generation process ensures that such optimization targets are always available for the training dataset and as such supervised model training is always possible. You are of course correct in that ground truths targets are not available ‘in science’, however, our approach does not rely on them for testing (or deployment).
>
> Secondly, we agree that there are ‘usually several equivalent ways to formally describe “laws of nature”’. Our skeleton recovery metric takes equivalent formulations of the same ODE into account, i.e. we do not compare predictions on the level of strings but on the level of mathematical expressions, as described in section 3.3 (last paragraph).
>
> Thirdly, regarding ‘[...] it is way more important to correctly recover the dynamical behavior supporting the system, i.e. the vector field and various of its topological and geometrical properties [...]’. We are very grateful for this comment as it helped us look at our problem setting from a different angle. We believe that your opinion is very reasonable and that there are probably many situations in which an exact symbolic expression may be less relevant than the qualitative properties of the dynamics. Although we did not consider this perspective during the course of our project, we believe that it does not contradict our problem statement or modeling approach: As described above, the data generation process ensures that supervised model training is always possible. Hence at test (or deployment) time, the trained model will be able to predict “a” (rather than “the”) symbolic expression to describe the observed data. Properties of the dynamics can then be deduced from the predicted equation. Hence predicting a symbolic expression is a reasonable (and sufficient) goal, even if we are “only” interested in e.g. geometrical properties.
>
> Fourthly, perhaps from a more practical perspective, evaluations on the symbolic level, that is, comparing the predicted vs “correct” (/expected) mathematical expression, is very common in the symbolic regression literature (see e.g. Qian et al. (2022) (section 5, Evaluation metrics.), Petersen et al. (2021) (section 4, last paragraph), Vastl et al. (2022) (section 4.6)). In that sense, we follow standard practice.
>
> Finally, we understand that reported performances do not appear impressive - however, as you rightly point out yourself, this is true for all models and hence also emphasizes the difficulty of the problem. We added an ablation experiment in appendix F to demonstrate that the performance is well-above a hypothetical random chance baseline.

---

> > ### Author Response · Authors · 2022-11-19
> > **Regarding your third comment**
> >
> > **3. There are other, more powerful approaches**
> >
> > Thank you for summarizing the goal of our paper and for acknowledging the principal value of our work. Also, thank you again for pointing us to the related literature and for providing a complementary view on interpretability. We have revised the related work section to reflect this view and to include additional references.
> >
> > Coming from the symbolic regression literature, Sindy appears limited in its expressiveness as it can not precisely represent nested or parametric symbolic functions unless these are explicitly included in its function library (which, for arbitrary functions, quickly leads to a combinatorial explosion.) Following your review we see that there are different viewpoints on expressiveness (e.g. approximation by polynomials) and have revised our statement in the manuscript.

---

> > > ### Comment · Reviewer_UR3m · 2022-11-24
> > > **further feedback**
> > >
> > > I much appreciate the authors’ replies and the broadening of the discussion in the paper to cover related areas. I think this is an important step to provide a more general context for the authors’ work. I’m aware that the issues I brought up are somewhat difficult to address, since they are more principled in nature and don’t provide a specific guideline on how to improve the current ms.
> > >
> > > Let me first comment on some of your replies:
> > >
> > > 1) The authors write “We agree that extensions beyond these assumptions are certainly interesting”, but in my mind they are actually imperative for scientific applications. There is really not that much happening in 1d, hardly any physics model of modern interest (not even a pendulum) that’s covered. And I doubt that extensions are that easily possible, since in 2d, and even more 3d or higher, you need to deal with a whole range of new dynamical possibilities where I can’t see the current approach to easily transfer \& scale. The problem is not so much that you can’t augment your system formally to accommodate additional equations (as laid out in Appx. A), the problem is that the inference problem will become disproportionately much harder (or impossible for certain systems). For this I don’t see a solution.
> > >
> > > 2) I don’t think I really misunderstood these points in my original reading, but the arguments still apply in my mind. For instance, I’m aware of course targets are only available during training, and you have this for your GT systems. My point though is that the objective of the training procedure may be misguided to begin with, by putting emphasis not onto mathematically important properties of the dynamical system but onto specific symbolic forms. For example, say the dynamical law is given by a $3^{rd}$ order polynomial. This may yield either 1, 2, or 3 equilibrium points in the dynamics, stable, unstable, or saddle. So you may recover the right functional form in principle, but this alone doesn’t guarantee at all you have captured these topologically most important properties (and in consequence, nothing important about the geometry either). But this is key for understanding dynamics!
> > >
> > > One thing the authors probably could have done (and I’m sorry this occurs to me only now, but perhaps it’s a good outcome of having this discussion): Examine in more depth whether, and under which conditions, important topological properties are preserved, i.e. location and type of equilib. points (with concrete graphical examples of curves from different initial conditions, GT \& reproduced). This may, potentially, have given some insight at least regarding the 2nd point.
> > >
> > > Whether the present work provides important and novel steps in the area of inference of dynamical laws in symbolic form I leave up to other referees to decide (and will be happy to go with their judgment).

---

> > > > ### Author Response · Authors · 2022-12-09
> > > > **Thank you for your feedback!**
> > > >
> > > > Thank you for your feedback, we definitely see the value in this discussion about the underpinning scientific value in our problem formulation and appreciate your perspective from a different scientific angle!
> > > >
> > > > However, we have the impression that the review score is not based on the originality and correctness of our work nor on the difficulty of our problem setting and its relevance to the ICLR community, but rather on 1) things we did not yet do (but also did not claim to be doing (systems of ODEs)) and 2) related, but different problem settings, which arguably may be of greater interest to certain sub-communities, 3) certainly interesting and insightful, yet non-established evaluation criteria (2. and 3. somewhat overlapping). E.g., the statement “Whether the present work provides important and novel steps in the area of inference of dynamical laws in symbolic form I leave up to other referees to decide (and will be happy to go with their judgment).”, seems somewhat inconsistent with your review score (reject, not good enough with confidence 4).
> > > >
> > > > We kindly ask the reviewer to re-evaluate whether the ranking of the actual contributions in our manuscript has been overshadowed by the mere existence of potentially even more interesting problem settings (to some communities), which we did not set out to tackle.

---

> > > > > ### Comment · Reviewer_UR3m · 2022-12-12
> > > > > **I'm not sure we are talking about *different* problem settings**
> > > > >
> > > > > The authors’ shifted a bit to a meta-level in their response. First, of course I can judge the significance of a paper only from my personal perspective and background (and actually I think a good review process should collect different angles). When I referred to other referees, what I meant to imply is that perhaps there’s something in here which is a significant step in symbolic processing, which I missed and cannot judge.
> > > > >
> > > > > My low rating is based on the fact that I feel this paper is not relevant for the purported addressee, science (even less so mathematics). In an autonomous 1st-order 1d ODE you can only have monotonic convergence to an EP or divergence to infinity (so Fig. 1, top-right, is actually misleading here!). Moreover, the type of evaluation I suggested is actually relatively straightforward to do, and it is not at all something exotic (interesting only for a particular sub-community) or non-established, but in fact textbook in dynamical systems. These are mathematically fundamentally important properties that need to be assured, in my mind, for the output to be of any use. Actually, as it stands, wouldn't one be much better off starting with SINDy with its large library of interpretable basis functions, and then simply convert the sparse representation it returns to tidy string output?

---

### Official Review · Reviewer_iM1D · 2022-10-23

**Confidence:** 3
**Clarity, Quality, Novelty And Reproducibility:** The work is clear and somewhat novel.
**Correctness:** 4
**Technical Novelty And Significance:** 3
**Empirical Novelty And Significance:** 3
**Recommendation:** 5

**Details Of Ethics Concerns:**

None.

**Strength And Weaknesses:**

Strengths:
1. The generation of symbolic models that use expressions commonly used by people to do analytical work is of tremendous importance.
2. The proposed pipeline, once trained, is much faster than other techniques.
Weaknesses:
1. The authors state the Neural-Symbolic Ordinary Differential Equation (NSODE) is a symbolic model generator, as opposed to Neural Ordinary Differential Equations (NODE) (Chen et al., 2018) that generate opaque models. However, both develop symbolic models. The only difference is that the NSODE causes ones that use functions that we people understand and are helpful for analytical work. But, from a definition point of view, both approaches produce equally symbolic models.
2. Even though the authors show that their "model performs better or on par with existing methods," it still cannot resolve most of the presented problems, signaling that this work is inconclusive.

**Summary Of The Paper:**

The authors present a Neural-Symbolic Ordinary Differential Equation (NSODE) that generates models straight out of the data. It is reported in the paper that once trained, the pipeline can produce models much faster than other techniques.

**Summary Of The Review:**

The authors present a Neural-Symbolic Ordinary Differential Equation (NSODE) that generates analytical models that are simpler to handle. The presented experiments show that it is as good as current techniques.

---

> ### Author Response · Authors · 2022-11-19
> **Thank you for your feedback, we will address both of your concerns below.**
>
> **1. Symbolic models**
>
> Thank you for your comment which raises the interesting philosophical question: when should a model be regarded as symbolic? We agree that one could regard Neural Ordinary Differential Equations (NODEs) (Chen et al. (2018)) as symbolic models since one can always write down the equation that a NODE implements, e.g. $\dot{y} = f(y) = (\sigma(w \cdot \sigma(...) + b)$. However, according to this definition it seems difficult to find any (digital) model that would not be labeled symbolic as one can in principle always write down the underlying formalism. In our manuscript we implicitly regarded a model to be symbolic if its primary goal is to identify and explicitly output a mathematical expression that describes the relationship between variables. This is a common (and mostly implicit) definition used throughout the symbolic regression literature. NSODE falls into this symbolic model category. NODEs on the other hand never explicitly reveal the mathematical expression that they implement. Moreover, we argue that even if the implemented expression was revealed, its complexity (assuming standard model architectures) would render it close-to-useless from an analytical perspective - that is, the model would remain opaque. NSODE on the other hand is designed to output a human readable mathematical expression. We revised the manuscript (section 2) to make this distinction more clear.
>
>
> **2. Performance**
>
> In our experience, scientific progress is a process where a single paper rarely gives the full solution or absolute answer, especially in machine learning. Collectively, the presented results in our manuscript demonstrate that symbolic regression for ODEs is rather challenging - and our work is a step towards solving this problem. We do not claim that our model achieves perfect performance nor that it cannot be improved in the future. On the other hand, we also want to emphasize that the presented results are well-above a random guessing baseline. We added an ablation experiment in appendix F to verify this.

---

> > ### Author Response · Authors · 2022-12-09
> > **Did our rebuttal address your questions?**
> >
> > Given the provided ratings of the correctness as well as your acknowledgement of the technical and empirical novelty and significance of our work, could we resolve the confusion about NODEs and address the reviewer's single concerns (of whether results are inconclusive) in our rebuttal? If so, we kindly ask the reviewer to consider adjusting their score accordingly since no other weaknesses or criticisms were raised.

---

### Official Review · Reviewer_qT9N · 2022-10-24

**Confidence:** 4
**Correctness:** 3
**Technical Novelty And Significance:** 2
**Empirical Novelty And Significance:** 2
**Recommendation:** 3

**Clarity, Quality, Novelty And Reproducibility:**

The paper is well written and reproducible. While it is somewhat novel it is not well positioned in the body of existing literature. Due to improper evaluation its quality can't be reliably judged.

**Strength And Weaknesses:**

Strengths:
- Self contained. Presents a model and an approach to generate sufficient training data.
- Comparative analysis.
- Comparison on multiple datasets

Weaknesses:
- Models only single dimensional ODEs.
- Doesn't include approaches to inferring ODEs in the related work.
- Comparison to non-ODE baselines.
- Biased evaluation datasets.


**Summary Of The Paper:**

The authors propose Neural Symbolic Ordinary Differential Equation (NSODE) a model mapping trajectories of numerical observations of the evolution of the behavior of a single variable through time to a symbolic expression that can generate those observations. They also present an approach to generating random symbolic expressions representing ODEs and simulations thereof starting from different initial conditions to train the model motivated by related work on symbolic regression for non-differential algebraic expressions.

**Summary Of The Review:**

The presented approach while well written and even promising, it is severely limited and not correctly evaluated. Therefore, I lean towards rejection at this time.

- Models only single dimensional ODEs.
While the authors claim to discover dynamical laws, attempting to generalize to the general domain of science, the scope of model dynamical systems is severely limited. Physical systems tend to be more complex and models of such systems used in a broad range of scientific disciplines are represented as systems of differential equations.
- Doesn't include approaches to inferring ODEs in the related work.
In the background and related work section the authors mostly discuss approaches to symbolic regression for non-differential algebraic expressions which address a different problem than the problem that the authors tackle with their approach. In the literature there exists a significant body of work addressing this problem. For example, Schmidt and Lipson 2009, or equation discovery for differential equations or process based modeling of dynamical systems. Furthermore, the only method mentioned in the related work claimed to address the problem of learning ODEs from data is NODE (Chan et al. 2018) which is in fact an approach to integrate (solve, simulate) differential equations.
- Comparison to non-ODE baselines.
Related to the previous point, the authors compare to symbolic regression approaches that have not been designed for learning differential equations and yet interestingly outperform the proposed approach both with regards to several performance metrics and execution time.
-Biased evaluation datasets.
The datasets that are generated using the approach can contain examples seen by the model and can be biased towards the presented approach. The main variability of examples in the dataset comes from the randomly selected initial conditions. Given the limitation to a single dimension, most of the equations are highly probable not to be sensitive to initial conditions and either converge to a single stable point including zero and infinity. Therefore multiple initial conditions can have very similar trajectories.

---

> ### Author Response · Authors · 2022-11-19
> **We thank the reviewer for his summary and comments which we will address point-by-point below.**
>
> **1. Models only single dimensional ODEs:**
>
> We agree with you that extensions beyond single dimensional ODEs are certainly interesting and relevant to make our approach applicable to a broader variety of problem settings. Although in section 3, Problem setting, we explicitly limit the scope of the main text to one-dimensional ODEs, we believe that this is not a fundamental limitation of the approach. In appendix A, section “Systems of equations”, we describe a conceptually straightforward way to implement an extension to systems of ODEs as well as extensions to other settings that we explicitly excluded in the problem definition before. An implementation and evaluation of these extensions is beyond the scope of the paper.
>
> **2. Doesn't include approaches to inferring ODEs in the related work:**
>
> We thank the reviewer for the pointers to additional related literature and have revised Section 2, Background and Related Work, to include them. We agree with the reviewer that our related work section includes models that were not originally developed for ODEs - however, we want to point out that the section is not limited to those (see e.g. Brunton et al. (2016), Weilbach et al. (2021), Atkinson et al. (2019), Chen et al. (2018), Liu et al. (2020), Long et al. (2020), Qian et al. (2022)). As we build on prior work in symbolic regression we want to provide an overview over different related approaches. The revised related work section makes the original goal of referenced models more clear.
>
> We are unsure about the comment on NODEs:
> ‘[...] the only method mentioned in the related work claimed to address the problem of learning ODEs from data is NODE (Chan et al. 2018) which is in fact an approach to integrate (solve, simulate) differential equations.’ From our understanding NODEs implement the function f of a differential equation y’ = f(y) using a neural network. Fitting a NODE to data involves integration (/solving ODEs) yet upon convergence, the model approximates f. In that sense it learns the ODE from data (in a black-box form). We would kindly ask the reviewer to clarify the purported issue with our statement.
>
> **3. Comparison to non-ODE baselines**
>
> Our baseline comparison includes Sindy which is originally intended for ODE inference (Sindy stands for “sparse identification of nonlinear dynamics”, Brunton et al. (2016)); as such the reviewer’s claim that we only compare to models that are not intended for ODE inference is incorrect. However, we are very grateful for raising this concern as it helped us realize that there is a potentially confusing statement in our manuscript, section 4.2 (Baselines, 3rd sentence): “​​First, no baseline is suited directly to infer dynamical laws, but only to infer functional relationships”. What we meant to say is that the baseline models all take a particular 2-step approach to symbolic regression: 1) fit a regression function f to capture the relationship y’(t) = f(y(t)), 2) once fitted to the data, read off the symbolic form of f. This is in contrast to our proposed model (NSODE) that predicts symbols directly. This makes a crucial difference as the 2-step procedure of the baseline approaches requires numerical regression targets y’(t) which are usually not observed. We have revised the manuscript to make this more clear.
>
> Beyond this potential confusion, the reviewer is correct in that our comparison with baselines also includes models that were not originally developed for ODE inference but for non-differential algebraic expressions (GPLearn, DSO, AIFeynman). Our choice of baseline comparison is largely informed by the availability of public implementations. As most previous work in the symbolic regression literature focuses on non-differential algebraic expressions, the number of available implementations for symbolic regression models for ODEs is limited. To include additional baseline models we hence turned to symbolic regression models for non-differential algebraic expressions and supplied them with finite difference approximations of temporal derivatives as regression targets as described in Section 4.2 (first paragraph). This is essentially the same strategy that Sindy (which was proposed for ODEs) uses to infer ODEs. Similarly, Weilbach et al. (2021) propose precisely this approach to use AIFeynman for ODE inference (but unfortunately, their implementation is not publicly available). Your pointer to the work from Schmidt & Lipton (2009) would have been a good additional comparison but the implementation does not seem to be publicly available either (see https://www.datarobot.com/nutonian/).

---

> > ### Author Response · Authors · 2022-11-19
> > **Regarding your comments on evaluation and correctness**
> >
> > **4. Biased evaluation datasets**
> >
> > We assemble different datasets to assess different levels of generalization. Assembly procedures and datasets are described in section 4.1 (Benchmark datasets). In short, these datasets assess:
> > Testset-iv: generalization to unseen initial values
> > Testset-constants: generalization to unseen initial values and constants
> > Testset-skeletons: generalization to unseen skeletons
> > Corresponding performance evaluations are found in Figure 2 as well as in Appendix F, table 8.
> >
> > We would kindly ask for clarification of the reviewer’s concern that “[t]he main variability of examples in the dataset comes from the randomly selected initial conditions”.
> >
> > If initial conditions were the main source of variability, the inference problem would become (even) more challenging as the symbolic expression would have to be (mostly) inferred from the initial condition rather than characteristic differences in trajectories. Since initial conditions are sampled i.i.d. across symbolic expressions (=samples), this would render the inference problem next to impossible.
> > However, the reviewer is of course correct in that a testset needs to be sufficiently diverse to assess generalization. As described in section 4.1, we ensure diversity by filtering out any duplicate skeletons in the testsets, that is, every sample in a testset corresponds to a unique skeleton. We therefore do not understand the reviewer’s claim that the main variability comes from the randomly sampled initial values.
> > Finally, we have added an ablation experiment in appendix F in the revised manuscript that demonstrates that samples are diverse.
> >
> > **5. Correctness**
> >
> > We believe that our revision made the paper more clear and would otherwise kindly ask the reviewer to point us to purportedly incorrect claims.

---

> > ### Comment · Reviewer_qT9N · 2022-11-21
> > **Response**
> >
> > I would like to thank the authors for the time and effort taken to respond to my comments and modify their paper accordingly. My evaluation scores will the same, with the exception of correctness which in light of the response will be modified.
> >
> > 1. It is unfortunate that the authors consider systems of ODEs beyond the scope of their work. In light of the related work, which includes approaches to modeling dynamical systems beyond one dimensional systems, a comment/conceptual solution on possible extension in the appendix does not improve the lack of novelty of the proposed approach. Especially if the claim is to applicability to discovery of dynamical laws.
> >
> > 2. The authors improved the related work section and now acknowledge more of the existing related work. To resolve the misunderstanding about NODE: NODE is a function approximator. While it can indeed be argued that it "learns" an ODE in an arbitrary form it does not give a concise closed-form expression to describe the dynamics. It definitely doesn't learn dynamical laws.
> >
> > 3. I agree the with the first part of the comment and appreciate the correction. For the second part, however, In light of the added related work beyond the seminal work by Schmidt and Lipton, the authors could have compared to some of the other suitable approaches. For example, some of the other evolutionary algorithm based approaches.
> >
> > 4. The limitation of considering a single dimensional ODE also limits the type of dynamical behavior that can be produced by the system, mainly in terms of the possible steady states. Therefore, the characteristic differences of the trajectories are mainly driven by the initial conditions and so their variance is limited to the initial time points. In the long term, the dynamical behavior of the ODEs will either "explode" or will be attracted to the same steady states. I would also expect that the steady states would be more dependent on the values of the constant parameters than the initial values. A system sensitive to the initial values would be chaotic system, which according to your response to the excellent point 1 of reviewer UR3m, is unfortunately also beyond the scope of your work.

---

> > > ### Author Response · Authors · 2022-12-09
> > > **Thank you for your comments! We will address them separately below.**
> > >
> > > 1)
> > > As detailed in the original manuscript as well as the discussion and further acknowledged by most reviewers, the novelty of our approach does not depend on whether it applies to systems of ODEs, but in how we methodologically address our core problem setting. Given its benefits over existing methods along the dimensions of interest in this work, falling short in another dimension does not constitute a lack of novelty. We also emphasize that the results of our model and all baselines demonstrate that even in the scalar case the problem is still challenging and far from being solved.
> > >
> > > 2)
> > > Thank you for the clarification. We share your understanding of NODEs in our manuscript (citing from Section 2, first paragraph):
> > >
> > >     *“Modeling dynamics and forecasting their behavior has a long history in machine learning. While NODEs (Chen et al., 2018) (with a large body of follow up work) are perhaps the most prominent approach, their inherent black-box character complicates scientific understanding of the observed phenomena. [...] we turn the focus to a different class of models in this paper and look at approaches that explicitly predict a mathematical expression in symbolic form that describes the observed dynamics.”*
> > >
> > >     Our original differences seem to stem from a semantic distinction of whether a “dynamical law” intrinsically needs to be symbolic or whether any (potentially blackbox) functional description of a gradient field is entitled to be called “dynamical law”. The diversity of reviewers’ comments on this point (reviewer iM1D considers NODEs to by symbolic; reviewer UR3m points us to many “black-box” (yet interpretable) models to recover dynamical laws) shows that different researchers may have different views on what it means to learn or recover a dynamic law. Instead of focussing on any particular viewpoint, the revised manuscript makes it clear to the reader in what way a particular model recovers or learns a dynamical law.
> > >
> > > 3)
> > > We chose a strong set of baselines that represent the qualitatively different existing approaches (including an evolutionary algorithm (GPLearn)) and pitched them against our method in a fair comparison. While one could of course always “add more baselines”, there is no reason to believe (also none provided in the reviews) that the results and overall findings would change.
> > >
> > > 4)
> > > Thank you for your comment, it has helped us understand your concern. We believe that there are two different aspects to disentangle here:
> > >
> > >     **4.1**)
> > >     As described in the manuscript and in our previous comment, every testset comprises many different functional forms of ODEs, e.g. a single testset could contain $\dot{y} = y$ and $\dot{y} = cos(y)$, etc.. With this in mind and given the difficulty of the task (i.e., considering the key criteria and requirements we set ourselves), scalar ODEs serve as an incredibly rich and difficult problem set. This is supported by our results in terms of the performance of the baselines and our method. Considering only the final qualitative result glances of the fact that solution trajectories of scalar ODEs can be arbitrarily rich before ultimately diverging or reaching a steady-state. As such solution trajectories for different functional forms will be inherently different. We therefore disagree with your statement that the variability in our dataset or the “characteristic behavior” of trajectories is driven by the initial value.
> > >
> > >     **4.2**)
> > >     Our dataset (=training + testing) contains multiple solution trajectories per functional form, e.g. multiple solution trajectories for the ODE $\dot{y} = y$. You are correct that given a particular functional form of an ODE (say: $\dot{y} = y$), solution trajectories for different initial values may be highly similar to each other (in non-chaotic systems). In fact, our supervised learning approach builds upon these similarities to infer the ODE expression for unseen trajectories at test time. However, importantly, the solution trajectory for each initial value is unique and we ensure that there is no overlap of initial values for a given functional form between training and testset. In other words, the model has not seen any of the trajectories of the testset during training. We hence disagree that our training- and testset setup biases the evaluation.
> > >     Concerning your statement “I would also expect that the steady states would be more dependent on the values of the constant parameters than the initial values.”, please consider our previous comment or the manuscript: We have a dedicated testset (testset-constants) which contains ODE expressions whose constants differ between training and test set.
> > >
> > >
> > > We kindly ask the reviewer to also read our last reply to reviewer UR3m and to also re-evaluate whether the ranking of the actual contributions in our manuscript has been overshadowed by the mere existence of potentially even more interesting problem settings (to some communities), which we did not set out to tackle.

---

### Author Response · Authors · 2022-11-19
**We thank all reviewers for their valuable feedback!**

Your comments helped us to improve our work. We have uploaded a revised manuscript, all text edits are marked in orange. We also re-aranged figures 2 and 3.

---

### Decision · Program_Chairs · 2023-01-20

**Decision:**

Reject

**Justification For Why Not Higher Score:**

The authors have to show the scalablity of the given approach to more complex cases (multi-D, higher order ODEs).

**Justification For Why Not Lower Score:**

N/A

**Metareview: Summary, Strengths And Weaknesses:**

The authors present a new method for extracting the symbolic expressions of an autonomous ODE from a time-series data stream. The paper only considers 1D first order ODEs, which is a quite limiting assumption. The architecture is based on a transformer.

The paper addresses a very hard and interesting problem However, the reviews for this paper was mixed. While reviewer LsD3 saw the paper very positive, all other reviewers had severe concerns regarding the scalability of the method as only 1D first order ODEs have been addressed. This excludes even the simplest physical dynamical systems such as the pendulum. Moreover there were concerns about the discussion of the related work and the choice of the baseline, which could be addressed mostly by the rebuttal. While the authors put a lot of effort in the rebuttal which also initiated at least to some extend a discussion with the reviewers, the reviewers stayed mainly negative due to the limitation to 1D first order systems. I agree that this is a severe limitation and scalability of the approach needs to be shown.

**Summary Of Ac-Reviewer Meeting:**

N/A